# Electron aspirator using electron–electron scattering in nanoscale silicon

Himma Firdaus[1], Tokinobu Watanabe[2], Masahiro Hori[1,2], Daniel Moraru[1,2], Yasuo Takahashi[3], Akira Fujiwara[4] & Yukinori Ono[1,2]

Current enhancement without increasing the input power is a critical issue to be pursued for electronic circuits. However, drivability of metal-oxide-semiconductor (MOS) transistors is limited by the source-injection current, and electrons that have passed through the source unavoidably waste their momentum to the phonon bath. Here, we propose the Si electron-aspirator, a nanometer-scaled MOS device with a T-shaped branch, to go beyond this limit. The device utilizes the hydrodynamic nature of electrons due to the electron–electron scattering, by which the injected hot electrons transfer their momentum to cold electrons before they relax with the phonon bath. This momentum transfer induces an electron flow from the grounded side terminal without additional power sources. The operation is demonstrated by observing the output-current enhancement by a factor of about 3 at 8 K, which reveals that the electron–electron scattering can govern the electron transport in nanometer-scaled MOS devices, and increase their effective drivability.

[1] Graduate School of Science and Technology, Shizuoka University, 3-5-1, Johoku, Naka-ku, Hamamatsu 432-8011, Japan. [2] Research Institute of Electronics, Shizuoka University, 3-5-1, Johoku, Naka-ku, Hamamatsu 432-8011, Japan. [3] Graduate School of Information Science and Technology, Hokkaido University, Sapporo 060-0814, Japan. [4] NTT Basic Research Laboratories, 3-1 Morinosato Wakamiya, Atsugi 243-0198, Japan. Correspondence and requests for materials should be addressed to Y.O. (email: ono.yukinori@shizuoka.ac.jp)

Electron–electron (e-e) scattering[1] conserves the total momentum of the electron system in semiconductors, and thus influences only indirectly the mobility and ON-state current of the transistors[2,3]. In fact, in state-of-the-art nano-transistors, the drivability is limited by the source injection current, which in turn is dominated by the momentum-nonconserving (MN) scattering processes involving interface roughness, impurities and phonons[4,5].

The e-e scattering, however, plays a crucial role in some particular conditions (and/or materials) when the e-e scattering length is shorter than that of the MN scattering. In such a case, hydrodynamic effects become apparent and some unusual phenomena, such as the absolute negative resistance[6] and higher-than-ballistic conduction[7], are observable.

Research studies of the e-e scattering and related hydrodynamic phenomena have been intensively performed at low temperatures in GaAs/AlGaAs heterostructures[8–14] and recently extended to other materials such as $PdCoO_2$ and graphene[15–17]. On the other hand, the research of the e-e scattering in Si MOS transistors is rather restricted to the magnetoconductance measurements[18–20] from the viewpoint of the carrier localization. Only one group has so far pointed out its importance for nano-scaled MOS transistors in their hot-electron experiments[21,22]. This stagnation may be because the MN processes, such as interface roughness scattering, have been believed to predominate in the electron transport in Si MOS channels[23–25]. However, since the e-e scattering itself is a momentum-conserving and energy-conserving process, the devices based on this process will guide us to a new concept for high-speed and low-power circuits, and thus research on this issue with Si, in particular on nanometer scale, is important. Actually, previous studies have investigated the hydrodynamic effects on micrometer-scale or larger, while the nano-scale hydrodynamics remains a frontier to be researched.

We thus propose a MOS device with a T-shaped branch to shed light on the role of the e-e scattering in nanometer-scale Si. We show that the device works as an "electron aspirator", in which energetic electrons from the inlet (emitter) induce a net electron flow from the grounded base terminal, resulting in the current enhancement at the outlet (collector).

The present result reveals that the e-e scattering has a crucial impact on the transport of nano-scaled Si devices, and that the hydrodynamic effects can be observed in such small devices. Most importantly, this is a key demonstration that one can increase the drivability of the MOS devices with negligibly small additional power dissipation. We also show that the interface roughness scattering, or what we call "the collision with the wall" in hydrodynamics, plays a significant role as a competing MN scattering against the e-e scattering.

## Results

**Device structure**. The device was fabricated on a silicon-on-insulator (SOI) substrate, and composed of emitter, collector and base. Figure 1a, b shows the schematic view of the device and a scanning electron microscope image. The emitter and collector have their own gates so that we can control the electrostatic potentials independently. The device also has a broad gate, referred here to as upper gate, which covers the emitter/collector gates and the T-branch region. By applying a positive voltage to it, electron inversion layers are formed beneath, which work as the electrical leads for the emitter-gate and collector-gate of the MOS transistors. The fabrication process[26,27] and the basic characteristics of the device can be found in the Method section and in Supplementary Fig. 1, respectively.

**Device characterization**. The experiments were performed at low temperatures (mainly at the substrate temperature $T = 8$ K), in which electrons were injected from the emitter to the T-branch region (Fig. 1c). In the experiments, the upper-gate voltage $V_{UG}$ and the substrate (or the back-gate) voltage $V_{SG}$ were fixed at positive and negative values (mainly 3.87 and –15 V), respectively, which ensures that the electron channels are formed only at the front interface of the SOI layer. With the above values for $V_{UG}$ and $V_{SG}$, the electron density $N$ and the Fermi energy $E_F$ at the T-branch region were estimated to be $5.7 \times 10^{12}$ cm$^{-2}$ and 36 meV, respectively. Except for the experiments for the last figure, the emitter was constant-current biased with the emitter current $I_E$ of –10 nA. Therefore, when we change the emitter gate voltage $V_{EG}$, the emitter voltage $V_E$ follows it and is auto-adjusted in order to keep $I_E$ constant. Unless otherwise mentioned, the collector and base currents, $I_C$ and $I_B$, were measured while keeping the collector and base terminals grounded.

Figure 1e shows $R_I$ as a function of the collector gate voltage $V_{CG}$, where $R_I = |I_C/I_E|$. We also show the log-scale plot of the $R_I$ in the inset of the figure. The voltage $V_{EG}$ was set at 0.7 to –1.8 V. These $V_{EG}$ values resulted in $V_E$ of –0.21 mV to –1.34 V, respectively. Note that these $R_I$ curves are identical to those of $I_C$ since $I_E$ is constant in the measurements. As one can see, when $V_{EG} = 0.7$ V, or $|V_E|$ is small (black curve), $R_I$ starts to increase at the threshold voltage of the collector gate $V_{CG-TH}$ ($\simeq 0$ V) and then becomes nearly constant at $R_I \simeq 0.35$–0.4. This strongly suggests that the collector and base are nearly equivalent from the viewpoint of the 'resistor' and the current flows diffusively (Fig. 1d left).

We should mention that this result for low $|V_E|$ is in striking contrast to the results of GaAs/AlGaAs lateral hot-electron transistors, where electrons travel to the collector nearly ballistically, which causes the current to flow below the threshold[28–31]. The present results indicate that the MN scattering is dominant for low $|V_E|$. Previous reports on the low-temperature transport of Si MOS transistors have revealed that the interface roughness and impurity (Coulomb) scatterings limit the mobility. (The phonon scattering has only a minor effect on the mobility at low temperatures.)[23–25] In fact, the low-temperature transport measurements of the present (undoped) SOI layer showed that the mobility was indeed limited by the interface roughness scattering for the front interface[32]. Therefore, we expect that the transport at the low-$|V_E|$ region is dominated by the interface roughness scattering.

When $V_E$ is negatively large (green to red curves), on the other hand, $R_I$ was found to significantly deviate from the low-$|V_E|$ case. $R_I$ exceeds unity ($I_B$ turns to negative, Fig. 1d right) and becomes as high as 3 for $V_{EG} = -1.8$ V. This is in effect the aspirator operation because the injection of energetic particles induces the net flow from the side channel. As shown in the inset of Fig. 1e, we observed a very small current below the threshold voltage of the collector gate, $V_{CG} \lesssim 0$ V. This is the current of hot electrons, i.e., of electrons whose energy is much larger than the $E_F$. As one can see, the hot-electron current is more than three orders of magnitude smaller than the main current above the threshold voltage. This indicates that the present current enhancement is not due to the hot electrons quasi-ballistically passing beneath the collector gate, but rather the small hot-electron current indicates the strong carrier scattering at the T-branch region.

Figure 1f shows the temperature dependence of the $R_I$. One can see that the $R_I$ enhancement is observed at temperatures up to about 100 K. Notice that we measured more than ten devices and found that all of them showed the enhancement effects though the maximum value of $R_I$ was scattered between about 1.5 and 3.5. (Notice that the performance improvement, in particular for the room-temperature operation, is discussed at the end of the main text and in Supplementary Note 5.)

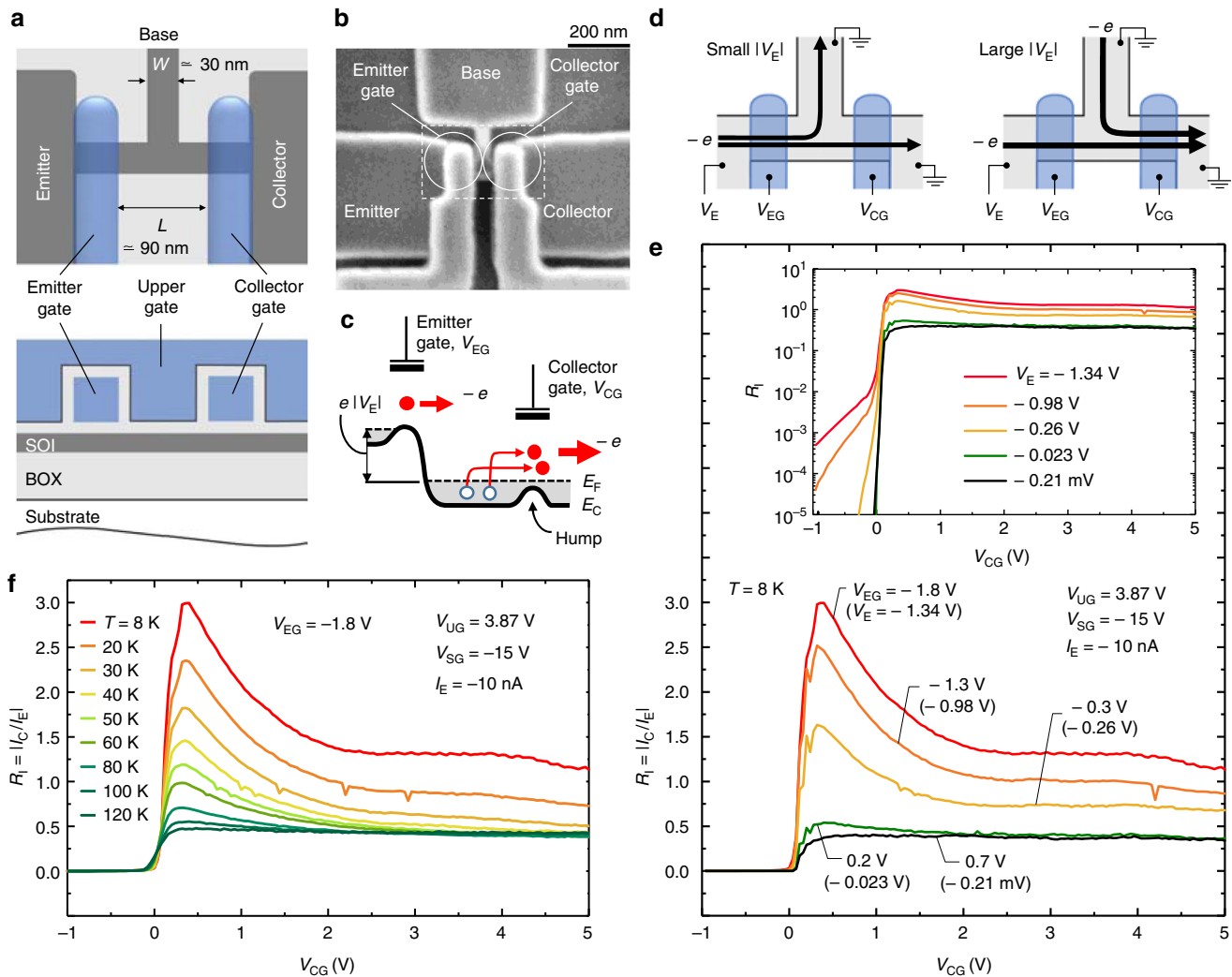

**Fig. 1** Structure and enhancement characteristics of the device. **a** Schematic top and cross-sectional views of the device. The front and back (buried) oxides, and the SOI are 20-nm, 390-nm, and 18-nm thick, respectively. The Si channel width $W$ in the T-branch region is about 30 nm. The length of the emitter and collector gates are about 60 nm and the spacing between the two gates $L$, which is an important parameter in the present study, is about 90 nm. **b** Scanning electron microscope image of a sample device with the same structure[26,27] as those measured here (before the upper-gate formation). Note that the Si channel and the gates are covered by oxide, and thus their actual widths are narrower than the appearance in the image. **c** Energy diagram of the device. $E_F$ and $E_C$ are the Fermi energy and the conduction band edge, respectively. Open blue dots represent holes. **d** Electron flow at small $|V_E|$ (left) and large $|V_E|$ (right). The arrows indicate the direction of the electron flow. **e** $R_I$ ( $= |I_C/I_E|$) as a function of the collector gate voltage $V_{CG}$ measured at 8 K. Inset shows the log-scale plot of the $R_I$. **f** $R_I$ as a function of $V_{CG}$ for the emitter-gate voltage $V_{EG} = -1.8$ V using the temperature $T$ as a parameter

Figure 2a shows the color plot of $R_I$ in the plane defined by $V_{CG}$ and $V_{EG}$. We also performed a different type of measurements, in which the base voltage $V_B$ was measured with the base terminal constant-current biased at $I_B = 0$ A. (The collector terminal was grounded.) In this case, $I_C + I_E = 0$ holds. (Note that, in this paper, the polarity of the current is defined by the direction of the electron flow in terms of the terminal, i.e., we define the current to be positive (negative) if electrons flow in (out of) the corresponding terminal.) Fig. 2b shows the color plot of $V_B$ in the plane defined by $V_{CG}$ and $V_{EG}$. One can see that, in spite of negative $V_E$, $V_B$ becomes positive, i.e., the resistance at the collector gate becomes negative. In other words, electrons flow towards the collector against the reversed bias. This is in effect the pumping operation, and the energy for the pumping is supplied from the emitter electrons. One can also see that the contour map of $V_B$ nearly coincides with that of $R_I$ in Fig. 2a.

What happens here is that, due to the $e$-$e$ collisions, the cold electrons at the T-branch region gain forward momentum,

forming an electron stream directed to the collector. The escape of electrons from the T-branch region leaves holes behind, which lowers the electrostatic potential there. (We mean by 'holes' positive charges created in the conduction band.) This potential drop attracts electrons, resulting in the electron flow from the base (Figs. 1e, 2a). When the base is current-biased at $I_B = 0$ A, this hole accumulation generates a positive $V_B$. In other words, this hole accumulation is compensated by a positive $V_B$ in order to hold $I_B = 0$ A (Fig. 2b). (In order for readers to understand how electrons move, we show expected potential profiles in the device in Supplementary Figs. 2, 3 of Supplementary Note 2.)

The phenomena can be observed when the scattering length $l_{ee}$ of the $e$-$e$ scattering is shorter than the device size (the spacing $L$ between the emitter and collector gates in the present case), $l_{ee} < L$. If the length $l_{MN}$ of the MN scattering is longer than the device size, $l_{MN} > L$, as well, the electron system can be treated as a "fluid" and the effect similar to the Venturi effect in hydrodynamics becomes effective[6,11–13]. As $l_{MN}$ becomes shorter,

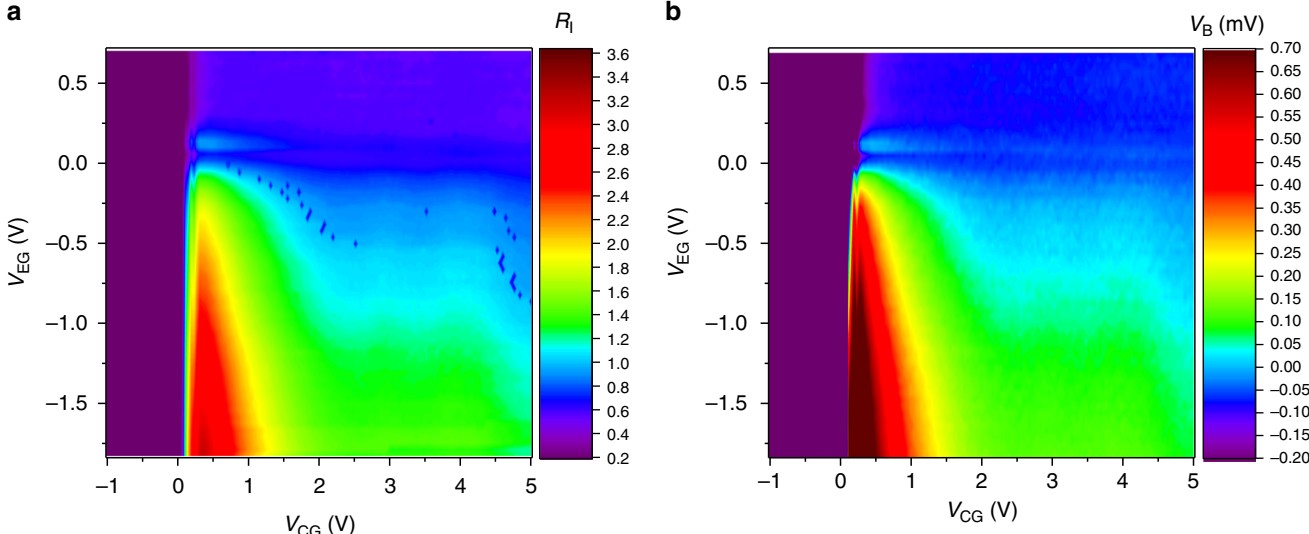

**Fig. 2** Color plot of the device characteristics in the $V_{CG}$ and $V_{EG}$ plane. **a** $R_I$ with the base terminal grounded ($V_B = 0$ V). Blue dots seen in the figure are the random telegraph noise presumably due to a single defect at the Si/SiO$_2$ interface. **b** $V_B$ with the base terminal current biased at $I_B = 0$ A

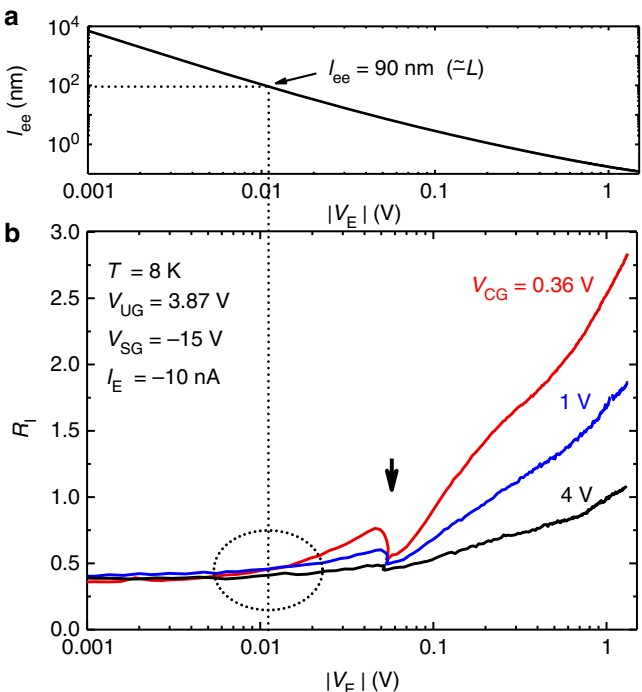

**Fig. 3** Calculated scattering length $l_{ee}$ and measured $R_I$ as a function of $|V_E|$. **a** Calculated scattering length $l_{ee}$. The vertical dotted line indicates the $|V_E|$ value ($=11$ mV) that gives $l_{ee} = 90$ nm ($\simeq L$). **b** $R_I$ as a function of $|V_E|$ using $V_{CG}$ as a parameter. The dotted circle shows the broad threshold at which the $R_I$ starts to increase. The arrow indicates the accidental drop of $R_I$

the directions of the momentum of each electron become more random (the total momentum of the electron system decreases), resulting in a reduced enhancement[33].

The presence of the electron fluid in the T-branch region can be claimed from the $R_I$ value for the large-$V_{CG}$ region. As shown in Fig. 1e, $R_I$ for large $|V_E|$ (e.g., the red curve) is larger than that for small $|V_E|$ (black) for the condition $V_{CG} \geq V_{UG}$ ($= 3.87$ V). This indicates that the total momentum of the electron system at the T-branch region is non-zero and directed to the collector. This is because, under this $V_{CG}$ condition, the potential hump at

the collector gate disappears, and thus the collector and base channels are equivalent from the viewpoint of the 'resistor'. Therefore, if the direction of the momentum is completely random in the T-branch region, $R_I$ should be equal to the low-$|V_E|$ value ($\simeq 0.35$–0.4), independent of $V_E$ (and thus of $V_{EG}$). This is not what we observed. (See also discussion around Fig. 5 for the claim that the directional momentum is present.)

If $V_{CG}$ is set at an appropriate value, the potential hump is formed (Fig. 1c) and it blocks the backflow of the low-energy ($\lesssim E_F$) collector electrons to be recombined with holes, making the aspirator more efficient (Figs. 1e and 2a). In other words, the collector gate works as a "check valve". Thus, we expect that the gradual decrease of $R_I$ as a function of $V_{CG}$ reflects the hole-density distribution below $E_F$.

The theory[34] based on the random phase approximation predicts that the e-e scattering rate $\tau_{ee}^{-1}$ due to the single particle excitation is given by

$$\frac{1}{\tau_{ee}} = -\frac{E_F}{4\pi\hbar}\left|\frac{\Delta}{E_F}\right|^2\left[\ln\left|\frac{\Delta}{E_F}\right| - \frac{1}{2} - \ln\left|\frac{2k_{TF}}{k_F}\right|\right] \quad (1)$$

where $\Delta = E - E_F$ is the electron energy with respect to $E_F$, and $k_F$ and $k_{TF}$ are Fermi and Thomas-Fermi wave numbers, respectively. This equation is applicable for $kT \ll \Delta \ll 2(k_{TF}/k_F)E_F$, which corresponds to 0.7 meV $\ll |eV_E| \ll 4$ eV, covering the present experimental conditions.

From the above formula, $l_{ee}$ can be estimated as $l_{ee} = v_{IN}\tau_{ee}$, where $v_{IN}$ is the initial forward velocity of electrons injected into the T-branch region. Figure 3a shows the calculated $l_{ee}$ as a function of $|V_E|$. (Details of the calculation can be found in Supplementary Note 3.) As indicated by the horizontal and vertical dotted lines, we obtain $l_{ee} = L$ ($\simeq 90$ nm) with $|V_E| \simeq 11$ mV. Larger $|V_E|$ makes $l_{ee}$ shorter, e.g., $l_{ee} \simeq 1$ nm at $|V_E| = 0.2$ V. For such a high $|V_E|$, the e-e scattering is expected to overwhelm the MN scattering due to the short $l_{ee}$. Thus, the injected electrons will experience multiple e-e scatterings immediately after entering the T-branch region, and transfer their forward momentum to many cold electrons until encountering the MN scattering.

Figure 3b shows $R_I$ as a function of $|V_E|$ using $V_{CG}$ as a parameter. A broad threshold indicated by the dotted circle can be seen at $|V_E| \sim 0.01$ V. This agrees with the calculated result, $l_{ee} = L$ at $|V_E| \simeq 11$ mV. Note that the dip indicated by the arrow was attributed to the trapping/detrapping of an electron to/from

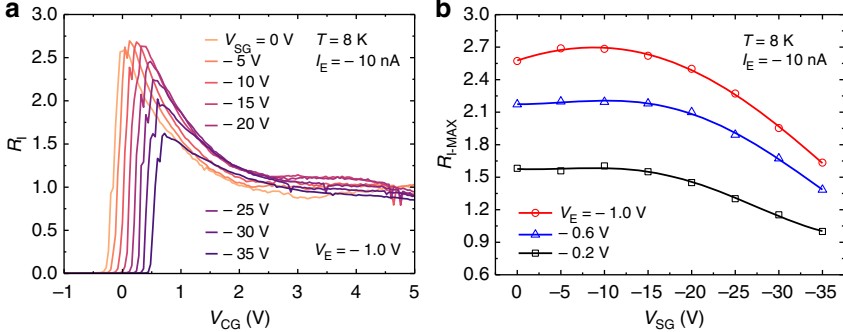

**Fig. 4** Substrate (back-gate) voltage $V_{SG}$ dependence of the device characteristics. **a** $R_I$ as a function of $V_{CG}$ using $V_{SG}$ as a parameter. **b** Maximum value $R_{I\text{-}MAX}$ of $R_I$ as a function of $V_{SG}$ using $V_E$ as a parameter

a localized state located near the emitter. This is not a common feature of the fabricated devices and its appearance is device-dependent. However, this is a strong implication that the momentum transfer property is drastically deteriorated by the Coulomb scattering, and that the present devices contain very few Coulomb scattering centers. Beside this accidental drop, $R_I$ monotonically increases with $|V_E|$.

We here notice that the previous Si MOS devices with similar structure did not provide the increase in the collector current, and the ratio of the collector to emitter currents was less than unity[21,22]. The critical difference between the previous and present devices is the channel doping concentration; undoped for the present devices, while boron-doped with a concentration on the order of $10^{18}$ cm$^{-3}$ for the previous ones. This implies that eliminating the impurity scattering is crucial for the present observations.

Then, the interface roughness scattering[35,36] is the most plausible candidate for the competing MN scattering process, if any. We therefore investigated the dependence of $R_I$ on the vertical electric field. Figure 4a shows $R_I$ as a function of $V_{SG}$. For these measurements, $V_{SG}$ and $V_{UG}$ were simultaneously varied so that the electron density $N$ at the T-branch region was kept constant ($= 5.7 \times 10^{12}$ cm$^{-2}$). Note that the vertical electric field is proportional to $V_{SG}$ for the triangular confinement potential of the SOI[32].

Figure 4b shows the maximum value $R_{I\text{-}MAX}$ of $R_I$ as a function of $V_{SG}$ using $V_E$ as a parameter. One can see that $R_{I\text{-}MAX}$ decreases with $V_{SG}$ (except when $V_{SG}$ is close to 0 V), supporting the idea that the interface roughness scattering plays a role. (A slight reduction observed at $V_{SG}$ close to zero may be due to the onset of the back channel at the buried interface which causes the inter-channel scattering[37,38].) Therefore, separating the channel from the interface by employing, e.g., the SOI volume conduction[39] and the use of the atomically flat Si/SiO$_2$ interfaces[40,41] will increase the enhancement capability.

From the viewpoint of hydrodynamics, the interface roughness scattering is nothing but the collision with the wall, whose strength determines the properties of the fluid[9,42]. In this sense, the electron fluid at the Si/SiO$_2$ interface will be an interesting experimental host for investigating how the friction with the wall affects the electron fluid.

In nano-scaled MOS transistors, a fraction of injected electrons turns back to the source due to the MN (back-) scattering, the amount of which determines the transferable momentum and thus the transistor drivability[4,5]. This means that any (MN or e-e) scatterings in the drain region play insignificant roles for the transistor ON-current[2,43], and electrons that have passed through the source merely waste their energy to the phonon bath. This energy dissipation process is unavoidable in the transistor. In the

electron aspirator, on the other hand, the channel electrons do work more efficiently than those in the transistor. They transfer their momentum to other electrons via the e-e scattering before they relax with the phonon bath, and this momentum transfer induces the base current, which flows between two grounded terminals, the base and the collector. This means that the electron aspirator enhances the transistor current with negligibly small additional power dissipation, which corresponds to the increase in the effective drivability of the device.

Figure 5a shows the transistor (gate-voltage) characteristics of the emitter, operating in the aspirator mode (the base is grounded) and in the transistor mode (the base is constant-current biased at $I_B = 0$ A). The voltage setup is shown in Fig. 5b. In these measurements, the emitter voltage $V_E$ (corresponding to the source voltage of the transistor) was kept at −1.0 V, while the collector (corresponding to the drain terminal of the transistor) was grounded. The voltage $V_{CG}$ was kept at 0.36 V. One can see that the aspirator-mode current (red solid curve) is larger than the transistor-mode current (red dotted curve) owing to the negative base current (blue solid curve) in the aspirator mode. This demonstrates that the present device can enhance the MOS transistor current with negligibly small additional power dissipation.

The ratio $R_{A/T}$ of the $I_C$ in the aspirator mode to $I_C$ in the transistor mode is also shown (black curve). Note that the $R_{A/T}$ is in effect the $R_I$. One can see that it is nearly constant at about 3 when $V_{EG} \lesssim$ −1.35 V, or the $I_C$ in the transistor mode (corresponding to $|I_E|$ for other figures) is less than about 3 nA (see dashed black lines in the figure). This is another evidence that the electron system at the T-branch region has a directional momentum. This is because the $R_{A/T}$ (i.e., $R_I$) is uniquely determined by $V_E$, not by $I_E$ or the injected power ($= |I_E V_E|$)[11,33].

In order to confirm that the device can operate with simpler voltage configuration, we also show in Fig. 5c the results for the case where the collector-gates and substrate-gates (back-gates) were both grounded. The voltage setup is shown in Fig. 5d. With this voltage setup, the device operates with only one supplemental gate (the upper gate) whose voltage is kept constant. One can see that the results are basically the same as those shown in Fig. 5a. (Note that the figures that include $I_E$ can be found in Supplementary Fig. 4 of the Supplementary Note 4.)

However, Figs. 5a, c reveal that the $R_{A/T}$ decreases when $I_C$ becomes larger than about 2–3 nA. We ascribed that the main cause of this degradation is the interface roughness scattering, which randomizes the direction of the momentum vector. Therefore, this problem will be relevant to the fact that the present aspirator works only at low temperatures. That is, another MN scattering, the phonon scattering, should be the cause of the momentum randomization at high temperatures.

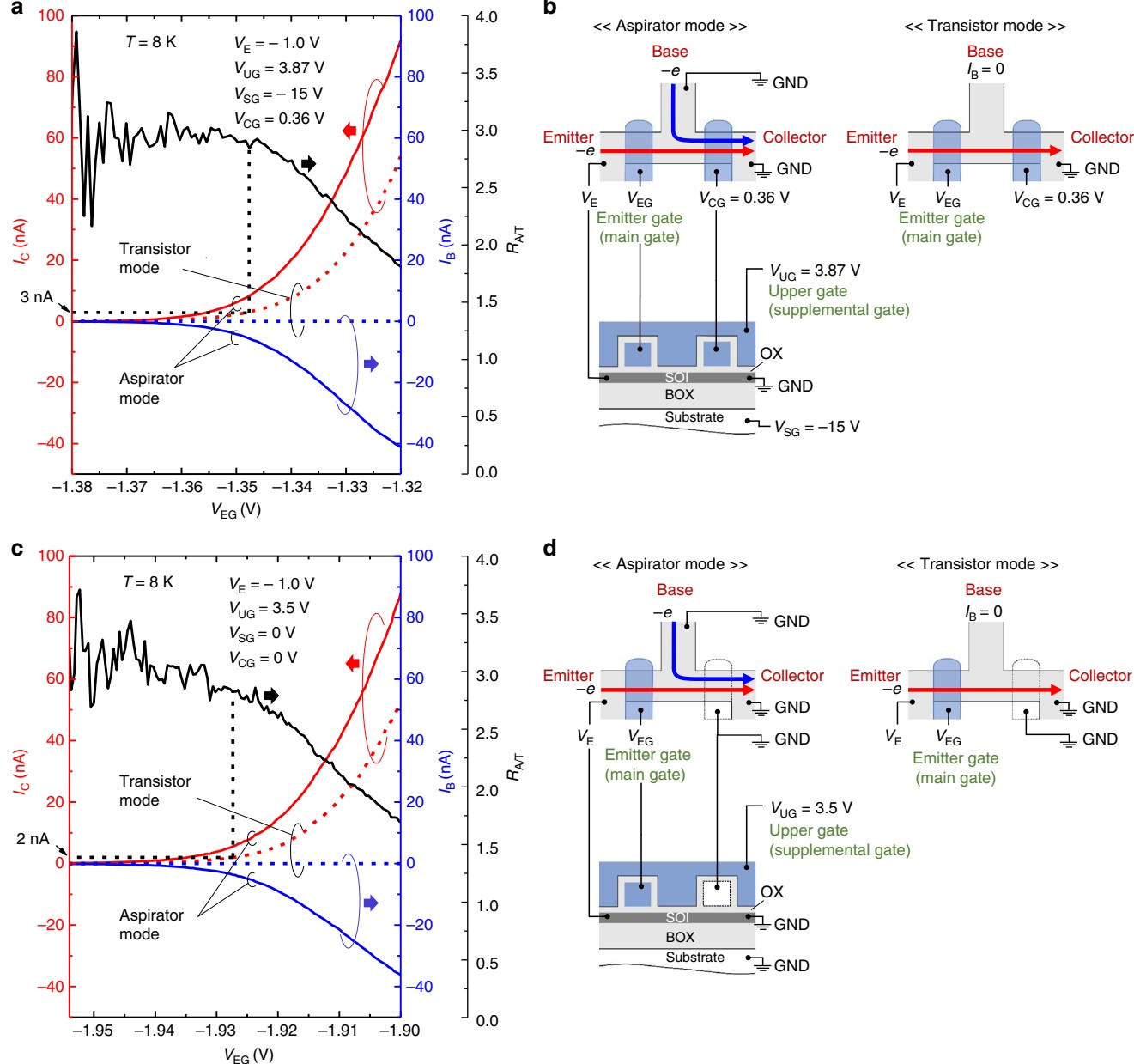

**Fig. 5** Comparison between the transistor-mode and aspirator-mode characteristics. The collector current $I_C$ and the base current $I_B$ as a function of the emitter gate voltage $V_{EG}$. The solid and dotted lines show the current curves in the aspirator-mode and transistor-mode, respectively. The collector (corresponding to the drain of the transistor) and the base currents are shown in red and blue, respectively. The $R_{A/T}$ is shown in black. The measurement setup for the aspirator- and transistor-modes are also shown. The arrows show the direction of the electron flow. **a** Measurement data with the collector and the substrate (back) gates being biased at $V_{CG} = 0.36$ V and $V_{SG} = -15$ V. **b** Voltage setup for data shown in **a**. **c** Measurement data with the collector and the substrate gates being grounded. **d** Voltage setup for data shown in **c**

## Discussion

Because of the above-mentioned problem, we systematically investigated the origin of the performance degradation ($R_I$ reduction) due to the high injection current and high temperature. Detailed arguments can be found in Supplementary Note 5, but here we briefly summarize the results. We investigated the temperature dependence of $R_I$ for various injection currents $I_E$. As a result, we found that the $R_I$ degradation at 8 K is not due to the increase in the lattice temperature $T$ caused by the high injection power, which in turn indicates that the cause is the other MN scattering, i.e., the interface roughness scattering. On the other hand, $R_I$ was found to decrease with the lattice temperature in a similar manner irrespective of the $I_E$ value, which strongly

suggests that $R_I$ is limited by the phonon scattering at high temperatures (see Supplementary Figs. 5–7).

Based on the above analysis, we propose here that the device is made smaller in order to satisfy the inequality $l_{ee} << L < l_{PH}$ (the phonon scattering length) for higher temperature operation. According to numerical simulations (Fig. 9 of ref.[44].), the electron-phonon scattering rate $\tau_{PH}^{-1}$ at room temperature is about 1 and $3 \times 10^{13}\,\mathrm{s}^{-1}$ for the electron kinetic energy of 0.1 and 0.3 eV, respectively. The electron-phonon scattering length $l_{PH}$ can then be estimated by using $l_{PH} = v_{IN}\tau_{PH}$, which results in $l_{PH} = 20$ and 30 nm for the electron kinetic energy of 0.1 and 0.3 eV, respectively. Therefore, the requirement for the room temperature operation will be $L \lesssim 20$–30 nm for the input voltage range $|V_E| = 0.1$–0.3 V.

The state-of-the-art nanotechnology has already enabled the fabrication of such small transistors, which we expect to raise the operation temperature of the present devices, as well.

We will finally estimate the maximum achievable value of $R_I$. In a scaled MOS transistor, the level of the current is determined by the source injection current[4,5], or by the forward momentum of an incident electron, $p_{IN}$, which can be expressed in the most simplified form as $p_{IN} = (2mkT)^{1/2}$ at room temperature (and as $p_{IN} = (2mE_F)^{1/2}$ at low temperatures), where $m$ is the effective mass. The injected electron is accelerated by the electric field in the channel and its momentum could reach $p = (2m(kT + |eV_E|))^{1/2}$. However, the momentum gained by the electric field, $p_{EL} \simeq (2m|eV_E|)^{1/2}$ (for $p_{EL} >> p_{IN}$) is unavoidably wasted to the phonon bath in MOS transistors. On the other hand, the aspirator uses the $p_{EL}$ for generating the base current, resulting in the output-current enhancement. If we could take a full use of it, the enhancement factor will reach $R_I \sim p/p_{IN} \simeq (|eV_E|/kT)^{1/2}$, or $R_I \simeq 3.5$ with $|V_E| = 0.3$ V at room temperature. Furthermore, if the injected electrons could scatter selectively with cold electrons with a forward momentum[11], further increase in $R_I$ may be possible. The estimation is encouraging, but detailed analysis about the dependence of $R_I$ on the device structure and the influence of the MN scattering will be called for to make any decisive conclusions. Exploring circuit architectures suitable for the device will also be needed.

There are several preceding reports regarding the electron hydrodynamic effects in solid-state devices and materials[8–16]. However, the device- and/or material-dimension was on the order of micrometer or larger. What we have demonstrated here is that the hydrodynamic effect can be observed and even predominate in the device characteristics on a nanometer-scale regime, where the energy dissipation issue is critical. The observation of the nano-hydrodynamic effect is mainly thanks to the heavy effective mass of electrons in Si, which gives us a short $e$-$e$ scattering length. A strong gate effect of Si MOS transistors, which enables us to generate a steep electric field, or a large voltage difference in a tiny area, will also be beneficial for the appearance of the hydrodynamic effect on nanometer-scale. Because of these material properties and well-developed Si nanotechnology, Si will be a suitable material for the research of the hydrodynamic effect on nano-scale.

In summary, the present results reveal that the $e$-$e$ scattering could govern the electron transport of nano-scaled Si MOS devices, and that the hydrodynamic effects can be observed in such small structures. In particular, we have shown that the Si electron nano-aspirator enhances the output current by a factor of about 3 at 8 K with negligibly small additional power dissipation, and thus increases the drivability of the MOS devices.

## Methods

**Device fabrication.** The device was fabricated on a (100) SIMOX (separation by implanted oxygen) substrate, a type of SOI, using the standard CMOS process, except for the use of electron-beam lithography for the Si wire formation. The SOI layer is undoped. All the gates are made of n$^{++}$ poly-Si. The fabrication was finalized with the annealing in H$_2$/N$_2$ ambient at 400 °C. This annealing effectively reduced the interface state density down to less than $1 \times 10^{10}$ cm$^{-2}$, which was confirmed by the charge-pumping current using large-area MOSFETs fabricated on the same wafer.

The fabrication process is the same as reported in refs. [26] and [27]. However, we employed thicker (18-nm thick) SOI layer and thinner (20-nm thick) gate oxide to minimize unwanted formation of Coulomb (single electron) islands due to potential fluctuation at the interface. Note that the stacked gate structure is an important feature of our device because the upper-gate driven electrical-lead formation prevents electrons from encountering the unwanted impurity scattering in the heavily-doped source/drain regions, allowing us to enhance the effects of the $e$-$e$ scattering process.

The channel width $W$ of the T-branch region and the spacing $L$ between the emitter and collector gates are, respectively, about 30 and 90 nm, as shown in Fig. 1(a).

**Electrical measurements.** The device was measured on a low-temperature probe station with the source monitor units (SMUs) of the semiconductor parameter analyzer Keysight B1500A. Special care has been taken for the offset of the input voltages generated by the SMUs. We measured the current flowing between the grounded base and collector terminals with the input emitter current $I_E = 0$ A, and confirmed that the offsets and fluctuation of the voltage between the base and collector terminals were negligibly small. Details of the measurements are shown in the Supplementary Note 6, with data shown in Supplementary Figs. 8, 9.

In this study, we mean by temperature $T$ the substrate (lattice) temperature, which was measured and calibrated by using a commercially available Si diode thermometer.

## Data availability

The authors declare that the data supporting the findings of this study are available within the paper and its supplementary information file.

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

## Acknowledgements

Y.O. thanks M. Tabe and K. Takashina for valuable comments and NTT Basic Research Laboratory for technical support. H.F. thanks RISET-Pro, Kemenristekdikti, Indonesia. This work was financially supported by JSPS KAKENHI (JP15H01706, JP16H04339, JP16H02339, JP16H06087, JP17H06211, and JP18H05258).

## Author contributions

A.F. and Y.T. fabricated the device. M.H. constructed the measurement system. H.F., T.W., M.H., and Y.O. measured the device. H.F., D.M., and Y.O. analyzed the data. Y.O. wrote the manuscript with support from H.F., D.M., A.F., and Y.T., and supervised the project. All the authors discussed the results.

## Additional information

**Competing interests:** The authors declare no competing interests.

