## [Peer Review file · Nature Communications]

Reviewers' comments:

Reviewer #1 (Remarks to the Author):

In this paper, the authors present a 3-terminal device where the collector current can be larger than the emitter one, even without biasing the third terminal. The current amplification is attributed to the so-called electron aspirator effect that relies on the properties of electron-electron interactions (energy and momentum conservation). The working principle is experimentally demonstrated at a temperature of 8K and supported by modeling considerations. It is expected that by reducing the device dimensions, the same effect could be observed at room temperature. The concept is novel and the demonstration is convincing, but the following minor points should be addressed before possible publication in Nature Communications:

-in Fig. 1e, what is shown by the IE and IC arrows does not seem to be the electrical current, but the electron flow. This should be corrected.

- at the top of page 5, the author claim that at low $|V_E|$, RI first increases and then stays constant, which "indicates that the collector and base are nearly equivalent from the viewpoint of the 'resistor' and the current flows diffusively". How can the conclusion be drawn that the current is diffusive? Assuming a perfectly ballistic current obeying the Landauer multi-terminal formula, one could also explain the observed current behavior. More justifications about the diffusive nature of the current should be provided.

- in between $V_E = -21$ mV and $V_E = -0.26$ V, some sort of turn-on happens, i.e. the current amplification at low VCG starts to take place. Is there a way to relate this turn-on to the applied voltages, the height of the potential barrier in the emitter and collector regions as well as the position of the Fermi level there?

- at the bottom of page 5, to get the condition $I_B = 0$, it is assumed that $I_C + I_E = 0$. It should rather be $I_C = I_E$, according to the direction of these flows in Fig. 1e.

- on page 6, it is written "this hole accumulation generates a positive V_B ". Wouldn't it be more accurate to say that "this hole accumulation must be compensated by a positive V_B "?

- on page 10, it can be read that "this demonstrates that the present device can enhance the drivability of the MOS transistor with negligibly small additional power dissipation". Why is the additional power dissipation negligible? If we assume that the dissipated power is $I_C * |V_{EC}|$, any increase of I_C by a factor of X leads to an increase in dissipated power by the same factor. Or how do the authors estimate the dissipated power?

- on page 11, how do the authors come to the estimation that the device length should be less than 20-30 nm for possible room temperature operation. Ref. 44 is provided for that, but the exact origin of the presented number is not clear.

Reviewer #2 (Remarks to the Author):

This paper report a new electron transport mechanism observed in Si metal-oxide-semiconductor transistors. Unlike the conventional mechanism that to switch on/off the device by changing the carrier concentration, it is shown that an injection current can scatter other electrons in the 3rd terminal to produce a current gain. The momentum-conserved collision

overwhelms the inelastic ones so as to display so-called viscosity of the electron flow. The authors call their devices electron aspirators. The hydrodynamical phenomena have been studied in materials of high mobility, such GaAs and 2D materials, such as graphene. There have been some trials in Si nanostructures/devices, however, the severe in-elastic scattering hindered the progress.

In this work, the collector current can be increased by 3 fold of the emitter current under a high emitter bias voltage when the temperature is lowered to 8 K. The authors also propose a possible dimension and working conditions for realizing the hydrodynamical Si device at room temperature.

The paper is interesting since it might be the first time to observe viscous electron transport in Si devices. However, the phenomenon is not very new and has been observed in many different systems. In particular, their device is very much similar to that used in REF 13. I would suggest the author may draw more lines why there will be an impact of their findings regarding science and technology issues.

My opinion is that the paper is publishable if the above concern and following minor issues are improved and explained.

The current gain appears when V_E is more negative, as one can see in Fig. 2(a) and Fig. 3. There seems no saturation at the largest $|V_E|$ the author used. I am curious about the trend if one can apply a more negative V_E than the current data.

All the data are converted into the I_C/I_E ratio. It is difficult to see how much current can be amplified in the optimal case.

What is the meaning of the blue dots in the color plot Fig. 2(a)?

Is the threshold for current gain (R_I) in Fig. 3 related to the threshold for current (I_E)? The application of theory Eq.(1) requires more careful inspection. Maybe the calculated mean free path for various V_E can be also shown in the same plot for the readers' better understanding your argument?

Can the data shown in Fig. 4 be explained by a change in Fermi energy (E_F) due to VSG? The ee scattering rate has E_F dependence.

The discussion on how to realize the aspirator at room temperature may be explained in further detail.

Reviewer #3 (Remarks to the Author):

The paper claims that they have fabricated a novel three-terminal MOS device utilizing e-e scattering which improves the drive current of the device at low temperatures. Unfortunately, I do not believe that the author has fabricated anything novel but instead the author has renamed standard nomenclature and improperly used others. There are many examples of this:

- 1.) The author claims the device is a three-terminal device when in fact the device requires 7 different power sources to operate
- 2.) The author claims that the device is a hot-electron transistor but does not explain how this device is a hot-electron transistor. This is counterintuitive as the definition of a HET is that the electron is injected from the emitter to the base and reaches the collector without scattering when the authors themselves are claiming that they are trying to enhance the drivability by utilizing e-e scattering
- 3.) The authors define their metric as R_1 which is classically known as alpha for the transistor. An alpha value of 1 considered infinite gain $\diamond I_B = 0$ and $I_E = I_C$. The authors show that in their device they are able to achieve an alpha of greater than 1 which means that something else is going on

(hence $I_B + I_E = I_C$) or the base is providing current to the collector. This means that this device is not operating in the fashion a hot-electron transistor would

4.) It is not clear from the paper what the doping concentrations are for this device at all. It appears that the collector, base, emitter are n-type ion implanted from the reference provided

5.) The device definitely requires an extra power source contrary to the claims of the author, the collector, base and emitter are all biased at different voltages and all of the gates consume power since power is require to charge and discharge the capacitors in the device. My intuition based upon what I have read in this paper is the base is simply another current provider and what the authors have done is effectively increased the width of the device. This device is in no way a practical way to save power as all of the gates need to be charged and discharged which consumes power and the improvement in drive current does not warrant such an increase in capacitance for switching.

Based upon my comments above, I think that the paper should be rejected for technical content and wording. The way that the paper is phrased is misleading, and claims of the authors are not correct based upon the ways they have worded it. The novelty of this paper is that it is a device with many gates, however, it provides no benefit to the actual performance of the device or to the way that people will design state-of-the-art electronics in the future

Responses to the Reviewers' comments

Reply to Reviewer #1:

Thank you for your valuable comments. According to your comments, we revised the manuscript. The revised parts are indicated by red characters in the manuscript. The following is the response to your comments.

In this paper, the authors present a 3-terminal device where the collector current can be larger than the emitter one, even without biasing the third terminal. The current amplification is attributed to the so-called electron aspirator effect that relies on the properties of electron-electron interactions (energy and momentum conservation). The working principle is experimentally demonstrated at a temperature of 8 K and supported by modeling considerations. It is expected that by reducing the device dimensions, the same effect could be observed at room temperature. The concept is novel and the demonstration is convincing, but the following minor points should be addressed before possible publication in Nature Communications:

- in Fig. 1e, what is shown by the I_E and I_C arrows does not seem to be the electrical current, but the electron flow. This should be corrected.

Thank you for your careful reading. In order to avoid the confusion, we deleted I_E and I_C from Figs. 1(c) and (d) in the revised manuscript, and added “- e” in place of them. We also added the explanation about the arrows in the figure caption.

- at the top of page 5, the author claim that at low $|V_E|$, R_1 first increases and then stays constant, which "indicates that the collector and base are nearly equivalent from the viewpoint of the 'resistor' and the current flows diffusively". How can the conclusion be drawn that the current is diffusive? Assuming a perfectly ballistic current obeying the Landauer multi-terminal formula, one could also explain the observed current behavior. More justifications about the diffusive nature of the current should be provided.

Thank you again for the careful reading. The branch in the present device is geometrically highly asymmetric with respect to the emitter; it is T-shaped with the emitter-collector path straight and the emitter-base path right-angled. Therefore, we expect that Landauer formalism predicts that the transmission probability to the collector is much higher than that to the base. Also, as we newly added in the inset of Fig. 1(e) of the revised manuscript, the log-scale plot of the R_1 demonstrates that the hot electron current, i.e., the current due to the electrons that have traveled quasi-ballistically, is very small. Therefore, we expect that the system is far from the conditions where the Landauer formalism can be applied.

However, as the Reviewer concerns, this expectation cannot be the direct demonstration of the diffusive transport. It might be difficult to demonstrate that the conductance is “perfectly” diffusive in the present device. Since it requires some elaborate measurements for this justification, and the justification of the diffusive transport in the low- $|V_E|$ region is not the main focus of the present study, we would like to decline our claim, and to use “strongly suggests” instead of “indicates”. We also used “expect” instead of “ascribe to” in a sentence on page 5. We think that this change does not affect any of our conclusions for the transport in the high- $|V_E|$ region.

- in between $V_E=-21$ mV and $V_E=-0.26$ V, some sort of turn-on happens, i.e. the current amplification at low V_{CG} starts to take place. Is there a way to relate this turn-on to the applied voltages, the height of the potential barrier in the emitter and collector regions as well as the position of the Fermi level there?

The turn-on threshold is determined by $|V_E|$, i.e., the Fermi level difference between the emitter and the T-branch region. This threshold property for the turn-on is independent of the emitter barrier height. (It simply controls the level of the injection current I_E .) Also, the turn-on threshold is independent of the collector barrier

height. (However, the value of $R_1 (= |I_C/I_E|)$ is dependent on the collector barrier height.) We will explain the turn-on below with some more details.

The turn-on, or the current enhancement, starts when the energy of the injected electrons exceeds a threshold value. Therefore, the turn-on is governed by the emitter voltage V_E . As indicated by Eq. (1), the e - e scattering time τ_{ee} , and thus the scattering length l_{ee} , becomes shorter when the injection energy increases (or $|V_E|$ increases). If $|V_E|$ is very low, the l_{ee} is longer than the length L of the T-branch region, $l_{ee} > L$. In such a case, the e - e scattering does not occur in the T-branch region, and thus no current enhancement occurs. In other words, the current enhancement occurs only when $l_{ee} < L$, for which we need a sufficiently large $|V_E|$ (or injection energy). Such threshold characteristics are shown in Fig. 3 of the main text.

Note that, according to the request by Reviewer #2, we added in Fig. 3(a) the calculated curve of the scattering length l_{ee} , evaluated from Eq. (1). As shown in the figure, a broad threshold can be seen at $|V_E|$ of about 10 mV (dotted circle), and this is in good agreement with the calculated results.

- at the bottom of page 5, to get the condition $I_B=0$, it is assumed that $I_C+I_E=0$. It should rather be $I_C=I_E$, according to the direction of these flows in Fig. 1e.

In the present manuscript, the polarity of the current is defined by the direction of the electron flow in terms of the terminal (just like what is used in semiconductor parameter analyzers). That is, we define the current to be positive (negative) if electrons flow in (out of) the corresponding terminal. In order to avoid the confusion, we added a sentence explaining this definition on page 6 in the revised manuscript. Note that preceding reports also took this definition.

- on page 6, it is written "this hole accumulation generates a positive V_B ". Wouldn't it be more accurate to say that "this hole accumulation must be compensated by a positive V_B "?

The accumulated holes charge the capacitor composed of the T-branch region, base channel and contact pad connected with the base channel. Then, the generated voltage due to the hole accumulation in this capacitor was sensed at the base terminal under the condition of $I_B = 0$. Therefore, in the experiment for Fig. 2(b), V_B is the sensed (measured) voltage, not the applied voltage. Therefore, we think that the present sentence ("This hole accumulation generates a positive V_B ") correctly describes the phenomenon.

However, as the Reviewer pointed out, this sentence could cause misunderstanding to the readers. Therefore, according to the Reviewer's suggestion, we made an additional explanation about this point on page 7.

- on page 10, it can be read that "this demonstrates that the present device can enhance the drivability of the MOS transistor with negligibly small additional power dissipation". Why is the additional power dissipation negligible? If we assume that the dissipated power is $I_C * |V_{EC}|$, any increase of I_C by a factor of X leads to an increase in dissipated power by the same factor. Or how do the authors estimate the dissipated power?

We appreciate such a fundamental question. This is indeed one of the important points of the paper. The collector current is composed of the emitter and base currents, i.e., $|I_C| = |I_E| + |I_B|$ in the case of the aspirator operation. The I_E is driven with the emitter-collector voltage V_E (or V_{EC} in your definition) while the I_B flows with the base-collector voltage, *which is definitely zero* (because both the collector and base are grounded). In other words, the base current flows under the zero bias condition. Therefore, the total power dissipation W is given by $W = |I_E V_{EC}| + |I_B \times 0| = |I_E V_{EC}|$, which is obviously less than $|I_C V_{EC}|$. This is why the aspirator can be an energy-efficient amplifier.

The real situation is a bit more complicated. The base electrons are introduced to the T-branch region by a very small potential drop δV_B whose value is less than 1 mV, as shown in Fig. 2(b), due to a finite value of the channel resistance. Then, such low-energy electrons arrived at the T-branch region gain the energy by the e - e scattering, which enables electrons to climb up to the collector against the small reverse voltage - δV_B . This is why we can obtain a finite value of the base current even under the zero bias condition. Therefore, a more accurate form for W is given by $W = |I_E V_{EC}| + |I_B \delta V_B|$, where $|\delta V_B| \ll |V_{EC}|$.

- on page 11, how do the authors come to the estimation that the device length should be less than 20-30 nm for possible room temperature operation. Ref. 44 is provided for that, but the exact origin of the presented number is not clear.

We used the data for the electron-phonon scattering rate shown in Fig. 9 of Ref. 44. This numerical simulation gives us the electron-phonon scattering rate τ_{PH}^{-1} of about 1 and $3 \times 10^{13} \text{ s}^{-1}$ for the electron kinetic energy of 0.1 and 0.3 eV, respectively. The electron-phonon scattering length l_{PH} was then estimated by using the simple formula, $l_{\text{PH}} = v_{\text{IN}} \tau_{\text{PH}}$, where v_{IN} is the forward injection velocity of electrons. How to calculate v_{IN} has already been described in the supplemental material. We added, on page 13 in the revised manuscript, some sentences in which we mention that we used Fig. 9 of Ref. 44 and v_{IN} for the estimation of l_{PH} .

Reply to Reviewer #2:

Thank you for your valuable comments. According to your comments, we revised the manuscript. The revised parts are indicated by blue characters in the manuscript. The following is the response to your comments.

This paper reports a new electron transport mechanism observed in Si metal-oxide-semiconductor transistors. Unlike the conventional mechanism that to switch on/off the device by changing the carrier concentration, it is shown that an injection current can scatter other electrons in the 3rd terminal to produce a current gain. The momentum-conserved collision overwhelms the inelastic ones so as to display so-called viscosity of the electron flow. The authors call their devices electron aspirators. The hydrodynamical phenomena have been studied in materials of high mobility, such as GaAs and 2D materials, such as graphene. There have been some trials in Si nanostructures/devices, however, the severe in-elastic scattering hindered the progress. In this work, the collector current can be increased by 3 fold of the emitter current under a high emitter bias voltage when the temperature is lowered to 8 K. The authors also propose a possible dimension and working conditions for realizing the hydrodynamical Si device at room temperature.

The paper is interesting since it might be the first time to observe viscous electron transport in Si devices. However, the phenomenon is not very new and has been observed in many different systems. In particular, their device is very much similar to that used in REF 13. I would suggest the author may draw more lines why there will be an impact of their findings regarding science and technology issues. My opinion is that the paper is publishable if the above concern and following minor issues are improved and explained.

The significance of the present work can be summarized as follows.

There are many preceding reports regarding the electron hydrodynamics in solid-state materials such as GaAs and graphene. However, all the material- and/or device-dimensions were on the order of micrometer or larger. The present work observes for the first time the hydrodynamic effect in Si, and demonstrates that it is effective *on a nanometer scale*. This has a crucial impact because it could lead us to apply the hydrodynamics to the circuits composed of highly scaled devices. The successful observation of the hydrodynamic effect on a nano-scale is mainly owing to the heavy effective mass of Si, which results in a very short $e-e$ scattering length, making it possible for the hydrodynamic effect to appear in such a small dimension.

In addition, we have demonstrated that the proposed nano-aspirator does enhance the injection current with negligible small additional energy dissipation (Fig. 5). This is the first demonstration that the electron hydrodynamic effect can increase the energy efficiency of electron devices. Since the energy dissipation is the most critical issue of the electronic circuits composed of devices with dimensions on the nanometer-scale, the present work is a significant step towards the *nano-scale electron hydrodynamics* for low power circuit architectures.

We will explain the above claim with a few more details.

1. Power of Si: Towards *Nano* hydrodynamics

As the Reviewer pointed out, our device is similar to that in Ref. 13. However, there is a critical difference, which is the device size; nanometer scale for ours, while micrometer scale for Ref. 13. This difference is not simply because of the lithography issue, but due to more fundamental reasons. The GaAs/AlGaAs device does not operate on nanometer-scale (or at least its performance is severely degraded.) This is mainly because of the effective mass of the material. Electrons in GaAs have a lighter mass than that in Si, which results in a longer $e-e$ scattering length l_{ee} . For the hydrodynamic nature of electrons to appear, the l_{ee} has to be shorter than any other scattering lengths and also than the device dimension. In addition, a strong gate effect of Si MOS transistors, which enables us to generate a steep electric field, or a large voltage difference in a tiny area, will also be important for the appearance of the hydrodynamic effect on nanometer-scale. These requirements are difficult to achieve in GaAs, and presumably in other materials, such as graphene, as well.

This problem is also closely related to the next Reviewer's question: "I am curious about the trend if one can apply a more negative V_E than the current data." As shown in Fig. R1 in this response letter, the amplification ratio R_1 monotonically increases with $|V_E|$ in our Si device. This is in striking contrast to the

GaAs/AlGaAs device in Ref. 13, where the hydrodynamic effect was lost for $|V_E|$ larger than about 0.3 V (Fig. 4 in Ref. 13). This is again because of the longer l_{ec} in GaAs.

The energy dissipation issue is important mostly in nano-scale devices in state-of-the-art electronic circuits. Then, we have discussed in the supplementary materials that a smaller device leads to higher temperature operation. We believe that Si is the best material solution for the device operating at a 10-nm scale, and thus being able to operate at room temperature.

2. Our proposal: Amplifying the transistor current by hydrodynamics

The core of our idea is the following. In the conventional transistor, a fraction of injected electrons turns back to the source due to the momentum-nonconserving back scattering (such as phonon and impurity scatterings), the amount of which determines the transferable momentum and thus the transistor drivability. The electrons that have passed through the source then contribute to the current, but they merely waste their energy to the phonon bath. *This energy dissipation process is unavoidable in the transistor.* In the electron aspirator, on the other hand, the channel electrons do work more efficiently than those in the transistor. They transfer their momentum to other electrons via the $e-e$ scattering before they relax with the phonon bath. This momentum transfer results in the deficiency of electrons at the drain (T-branch region) and it is compensated by the electron supply from the grounded base terminal with negligibly small power dissipation. Figure 5 is the experimental demonstration for this claim.

Let me explain why this device is energy-efficient by giving a few more details. The collector current is composed of the emitter and base currents, i.e., $|I_C| = |I_E| + |I_B|$ in the case of the aspirator operation. The I_E is driven with the emitter-collector voltage V_E , while the I_B flows with the base-collector voltage, *which is definitely zero* (because both the collector and base are grounded). In other words, the base current flows under the zero bias condition. Therefore, the total power dissipation W is given by $W = |I_E V_E| + |I_B \times 0| = |I_E V_E|$, which is obviously less than $|I_C V_E|$. The real situation is a bit more complicated. The base electrons are introduced to the T-branch region by a very small potential drop δV_B whose value is less than 1 mV, as shown in Fig. 2(b), due to a finite value of the channel resistance. Then, such low-energy electrons arrived at the T-branch region gain the energy by the $e-e$ scattering, which enables electrons to climb up to the collector against the small reverse voltage - δV_B . This is why we can obtain the finite value of the base current even under the zero bias condition. Therefore, a more accurate form for W is given by $W = |I_E V_E| + |I_B \delta V_B|$, where $|\delta V_B| \ll |V_E|$.

We also mention that the purpose of the preceding reports (refs. 21 and 22) on Si hot-electron transistors was not the observation of the hydrodynamic effects, but merely the observation of the ballistic electrons in Si. No attempts for the observation of the hydrodynamic effects in Si have been made so far. We think that this stagnation is because no one recognized that the hydrodynamic effects are fundamentally important from viewpoint of the energy efficiency in electronic circuits, and the present work is the first one that experimentally demonstrates this significance.

As stated above, we believe that the present work is significant from the viewpoint of science and technology in the field of nano-electronics. In the original manuscript, the description of the significant points was too conceptual. Therefore, we largely revised the abstract and the summary. In particular, we did not make a detailed explanation either why Si has an advantage over other materials, or why the present device is an energy-efficient one. Therefore, in the revised manuscript, we describe these points on pages 10 – 11 and at the end of the main text on page 13. We also added one sentence in the Introduction part (page 2) in order to emphasize the importance of the nano-scale hydrodynamics.

The current gain appears when V_E is more negative, as one can see in Fig. 2(a) and Fig. 3. There seems no saturation at the largest $|V_E|$ the author used. I am curious about the trend if one can apply a more negative V_E than the current data.

Please find Fig. R1 shown below. This is Fig. 3(b) of the revised manuscript, but the horizontal axis is changed to the linear scale. One can see that no saturation is observed. As we have explained above, this is an advantage of the Si device. In an ideal case, the increase in the amplification $R_1 - R_1 (V_E = 0)$ will be proportional to the forward momentum p of an injected electron, which is given by $p = (2m(|eV_E| + E_F))^{1/2} \simeq (2m|eV_E|)^{1/2}$ (for $|eV_E| \gg E_F$). That is, $R_1 - R_1 (V_E = 0) \propto |V_E|^{1/2}$. The observed dependence of R_1 on $|V_E|$ can be

explained by this simple formula. Therefore, larger $|V_E|$ should result in larger R_I . However, we cannot apply $|V_E|$ larger than about 1.5 V because the application of such a high voltage could cause the gate oxide breakdown in the present device.

Fig. R1. Dependence of R_I on $|V_E|$. The arrow shows the accidental drop of R_I .

All the data are converted into the I_C/I_E ratio. It is difficult to see how much current can be amplified in the optimal case.

In Fig. 5, we have shown the collector and base currents, instead of the ratio $R_I = |I_C/I_E|$. As one can see, currents on the order of 100 nA can be amplified. It is possible to amplify even higher currents, but the amplification capability is degraded. As it has been discussed in the supplemental material, we expect that this is due to the interface roughness scattering and that the scaling down of the device will improve this degradation.

What is the meaning of the blue dots in the color plot Fig. 2(a)?

The blue dots are caused by the random telegraph noise (RTN), presumably due to a single defect located at the Si/SiO₂ interface. The RTN can be also seen in some of the R_I curves in Fig. 1(e), and in Fig. 1(f) of the revised manuscript. However, this RTN does not affect any of our conclusions. We explain these dots in the figure caption of Fig. 2.

Is the threshold for current gain (R_I) in Fig. 3 related to the threshold for current (I_E)? The application of theory Eq. (1) requires more careful inspection. Maybe the calculated mean free path for various V_E can be also shown in the same plot for the readers' better understanding your argument?

Thank you for your suggestion. According to the Reviewer's suggestion, we added the calculated results of the mean free path (i.e., the scattering length l_{ec}) based on Eq. (1) in Fig. 3(a). (The detail of the calculation has already been described in the supplemental materials.) As shown in the calculated results, the l_{ec} monotonically decreases with increasing $|V_E|$. The R_I should start to increase when the scattering length l_{ec} becomes shorter than the device size (length L between the emitter and collector gates), $l_{ec} < L$. As shown in Fig. 3, this prediction is in good agreement with the experimentally observed threshold voltage. According to this revision, we added a few words for explaining the calculations on page 8.

We mention that, in all the experiments, except for that for Fig. 5, we used a constant emitter current I_E . This means that the emitter transistor was always in ON state. Therefore, the threshold of the R_I is not relevant to the threshold of the emitter current I_E .

Can the data shown in Fig. 4 be explained by a change in Fermi energy (E_F) due to V_{SG} ? The $e-e$ scattering rate has E_F dependence.

As the Reviewer points out, the $e-e$ scattering rate is dependent on E_F . In other words, it is dependent on the electron density N at the T-branch region. Therefore, we kept the electron density N at the T-branch region constant at $N = 5.7 \times 10^{12} \text{ cm}^{-2}$ in the experiment for Fig. 4. In other words, we kept the E_F constant at 36 meV. For this purpose, we adjusted the upper-gate voltage V_{UG} in accordance with the change ΔV_{SG} in V_{SG} , as $\Delta V_{UG}/\Delta V_{SG} = \alpha$ ($= -0.03$), where $|\alpha|$ is the ratio of the upper gate capacitance to the substrate (back) gate capacitance. This experimental setup has already been described in the supplemental material and also touched on in the main text. (The corresponding part is underlined on page 9 of the main text.)

The discussion on how to realize the aspirator at room temperature may be explained in further detail.

Thank you for the suggestion. According to the Reviewer's suggestion, we added some sentences on pages 12 – 13. These are the summary of the analysis and discussion made in the supplemental material.

Reply to Reviewer #3:

Thank you for your valuable comments. According to your comments, we revised the manuscript. The revised parts are indicated by green characters in the manuscript and in the supplemental material. The following is the response to your comments.

The paper claims that they have fabricated a novel three-terminal MOS device utilizing e - e scattering which improves the drive current of the device at low temperatures. Unfortunately, I do not believe that the author has fabricated anything novel but instead the author has renamed standard nomenclature and improperly used others. There are many examples of this:

I am afraid that the criticisms raised by the Reviewer are due to the insufficient data and explanation about how the device operates, and also due to our misleading writing of the manuscript. Therefore, we prepared additional data in order to answer to the Reviewer's concern about the multi-power-supply configurations. We also deleted the terms that cause the misunderstanding to the readers, such as "three-terminal", from the manuscript.

We show below the answers to your questions 1) – 5).

1.) The author claims the device is a three-terminal device when in fact the device requires 7 different power sources to operate.

As the Reviewer points out, the present device is not a three-terminal device. Therefore, we deleted the term "three-terminal" from the manuscript.

However, we mention that the multi-terminal configuration does not destroy the low-power nature of the present device. The gate terminal that has to be charged and discharged, and thus consumes the power is only the emitter gate. The other gates are not relevant to the energy dissipation because their voltages are kept constant for the device operation (Fig. 5). More importantly, the present device can operate with much simpler voltage configuration. Detailed explanation on this point will be made in the Answer to the Reviewer comment 5).

2.) The author claims that the device is a hot-electron transistor but does not explain how this device is a hot-electron transistor. This is counterintuitive as the definition of a HET is that the electron is injected from the emitter to the base and reaches the collector without scattering when the authors themselves are claiming that they are trying to enhance the drivability by utilizing e - e scattering.

Thank you for your comments. As the Reviewer pointed out, the term of "hot-electron transistor" is misleading because the observed hydrodynamic effect is not relevant to the hot-electron transistor effect. Therefore, we decline to use this terminology for the present device.

However, we mention that hot electrons are indeed present in the channel. Figure R2 shows the log-scale plot of R_1 ($=|I_C/I_E|$), which corresponds to Fig. 1(d) of the original manuscript (and to Fig. 1(e) in the revised manuscript). Notice that the R_1 is in effect $|I_C|$ because I_E was kept constant ($= -10$ nA) in this experiment. One can see that the tails are observed in the range of $V_{CG} \lesssim 0$ V (indicated by dotted circle). This is due to the current of hot electrons, i.e., of electrons passing beneath the collector gate with a much higher energy than the Fermi energy. One can also see that the level of the current due to the hot electrons is very low, more than three orders of magnitude lower than the main current components above the threshold voltage ($V_{CG} \gtrsim 0$ V). This is in striking contrast to the GaAs/AlGaAs counterparts, where the hot electrons can predominate in the conductance. We expect that this low level of the hot-electron current is because the injected electrons lose their energy by the e - e scattering, and this in turn supports the idea that the hydrodynamic effect is quite strong in the present device.

We added this figure (Fig. R2) in the inset of Fig. 1(e) in the revised manuscript. By this addition, we hope that one can make a deeper understanding of how the present device operates and how the hydrodynamic effect

emerges there without any confusion. According to this revision, we also added some sentences explaining the hot-electron component on page 5 in the revised manuscript.

Fig. R2. Log-scale plot of $R_1 (= |I_C/I_E|)$ as a function of the collector gate voltage V_{CG} . The hot electron current is observed in the V_{CG} region below the threshold voltage ($V_{CG} \lesssim 0$ V).

3.) The authors define their metric as R_1 which is classically known as alpha for the transistor. An alpha value of 1 considered infinite gain $I_B = 0$ and $I_E = I_C$. The authors show that in their device they are able to achieve an alpha of greater than 1 which means that something else is going on (hence $I_B + I_E = I_C$) or the base is providing current to the collector. This means that this device is not operating in the fashion a hot-electron transistor would.

As it has been explained above, the observed hydrodynamic effect is not due to the hot electrons quasi-ballistically passing beneath the collector gate, or is not relevant to the hot-electron transistor operation. Therefore, we declined to use the term of “hot-electron transistor” for the present device.

4.) It is not clear from the paper what the doping concentrations are for this device at all. It appears that the collector, base, emitter are n-type ion implanted from the reference provided.

As the Reviewer pointed out, the device has n-type ion implanted regions, and the doping density in these regions is higher than 10^{20} cm^{-2} . However, these are located outside the upper-gate region. For the fabrication of the device, the ion-implantation was performed using the upper gate as a mask in a self-aligned manner. Therefore, the core parts of the device, i.e., the emitter and collector channels and the T-branch region, remain undoped (nominally the doping concentration less than 10^{15} cm^{-2}) because they are located beneath the upper gate.

As we have stated in the “Device Structure” section, by applying a positive voltage to the upper gate, electron inversion layers are formed in the T-branch region, which work as the electrical leads for the emitter and collector. This is why the emitter and collector can work without being attached to the heavily-doped regions.

It is also mentioned that, although the device structure is similar to the one in Refs. 26 and 27, the fabrication process has been improved for the present work. We used undoped and thicker SOI layer with thinner gate oxide as compared to the one in Refs. 26 and 27. This point has been mentioned in the Methods section of the manuscript.

5.) The device definitely requires an extra power source contrary to the claims of the author, the collector, base and emitter are all biased at different voltages and all of the gates consume power since power is require to charge and discharge the capacitors in the device. My intuition based upon what I have read in this paper is the base is simply another current provider and what the authors have done is effectively increased the width of the device. This device is in no way a practical way to save power as all of the gates need to be charged and discharged which consumes power and the improvement in drive current does not warrant such an increase in capacitance for switching.

Multi-power-source configuration

In the present device, the gate terminal that has to be charged and discharged, and thus consumes the power, is only the emitter gate. The other gates (the collector-, upper-, and substrate-gates) are not relevant to the energy dissipation because their voltages are kept constant for the device operation shown in Fig. 5. We also mention that the back and collector gates are the control gates. These gates were necessary for Figs. 1 – 4 to vary the parameters such as electron density, to clarify the physics behind the device operation. However, biasing to these gates is not necessary for the practical device operation shown in Fig. 5. In fact, the device can operate with these gates being grounded.

Figure R3 shown below is the experimental data (corresponding to Fig. 5 in the manuscript), for which the collector- and substrate-gates, as well as the collector and base, were grounded. One can see that the device can still properly operate. The voltage to the upper gate is the only supplemental power source (but this voltage is kept constant) in this device operation. (The main gate is the emitter gate.)

For the fabricated device, biasing to the upper gate is necessary for the electron channel to be formed in the T-branch region. However, we expect that the upper gate can also be grounded if we take the metal gate and appropriately select the material so that the channel is formed automatically due to the work-function difference between the metal and Si. In such a case, the device requires only one power source and the ground.

We replaced Fig. 5 with this figure (Fig. R3), and added some sentences on page 11. The original Fig. 5 was moved to the supplemental material as Fig. S2.

Fig. R3. Comparison between the transistor- and aspirator-mode characteristics, corresponding to Fig. 5 of the manuscript. For this measurement, the collector, base, collector gate, and substrate (back gate) were grounded. The upper gate is the only supplemental gate, but its voltage is kept constant.

Energy dissipation in the device

We would like to claim that the present device definitely enhances the current with negligibly small additional power dissipation, and this is quite fundamental. (The device is not merely a transistor with a wider channel.)

In the MOS transistor, a fraction of injected electrons turn back to the source due to the momentum-nonconserving back scattering (such as phonon and impurity scatterings), the amount of which determines the transferable momentum to the channel. The electrons that have passed through the source then contribute to the current, but they merely waste their energy to the phonon bath. *This energy dissipation process is unavoidable in the transistor.* In the present device, on the other hand, the channel electrons do work more efficiently than those in the transistor. They transfer their momentum to other electrons via the $e-e$ scattering before they relax with the phonon bath. This momentum transfer results in the deficiency of electrons at the drain (T-branch region) and it is compensated by the electron supply from the grounded base terminal with negligibly small power dissipation.

Let me explain why this device is energy-efficient by giving a few more details. The collector current is composed of the emitter and base currents, i.e., $|I_C| = |I_E| + |I_B|$ in the case of the aspirator operation. The I_E is driven with the emitter-collector voltage V_E , while the I_B flows with the base-collector voltage, *which is definitely zero* (because both the collector and base are grounded). In other words, the base current flows under the zero bias condition. Therefore, the total power dissipation W is given by $W = |I_E V_E| + |I_B \times 0| = |I_E V_E|$, which is obviously less than $|I_C V_E|$.

The real situation is a bit more complicated. The base electrons are introduced to the T-branch region by a very small potential drop δV_B whose value is less than 1 mV, as shown in Fig. 2(b), due to a finite value of the channel resistance. Then, such low-energy electrons arrived at the T-branch region gain the energy by the $e-e$ scattering, which enables electrons to climb up to the collector against the small reverse voltage $-\delta V_B$. This is why we can obtain the finite value of the base current even under the zero bias condition. Therefore, a more accurate form for W is given by $W = |I_E V_E| + |I_B \delta V_B|$, where $|\delta V_B| \ll |V_E|$.

Based upon my comments above, I think that the paper should be rejected for technical content and wording. The way that the paper is phrased is misleading, and claims of the authors are not correct based upon the ways they have worded it. The novelty of this paper is that it is a device with many gates, however, it provides no benefit to the actual performance of the device or to the way that people will design state-of-the-art electronics in the future.

As we have explained, we removed the misleading terms from the manuscript. Also, we have shown that the hydrodynamic effect due to the $e-e$ scattering can enhance the current with negligibly small additional power dissipation, and that the present device can operate with much simpler voltage configuration than that we originally showed.

On behalf of the authors, I would appreciate if you could accept our response, read the revised manuscript, and reconsider the possibility of the publication.

Thank you for your kind consideration.

Reviewers' comments:

Reviewer #1 (Remarks to the Author):

The authors carefully addressed all my comments. I'm therefore in favor of accepting this manuscript.

Reviewer #2 (Remarks to the Author):

The authors have made a substantial revision to the points I raised. I recommend the acceptance for publication.

Reviewer #3 (Remarks to the Author):

I thank the authors who have put out the effort to answer the questions of the reviewers. With the additional clarifications, it is clearer now than ever that there is something strange going on. Figure 3d shows that in aspirator mode, there is current flowing from the base which is grounded to the collector which is also grounded. This to me signifies that there is something going on where one or both of the terminals is not truly held at ground. In figure 5, shows that the current to the base and collector are of opposite polarities giving the authors the "amplification" that they speak of in this paper. I would like to point out that if both the base and collector are sitting at ground, then the polarities of these currents must be the same otherwise the node rule of Kirchoff's current laws is violated. The common node between the base and collector is the intersection point of the T where the potential is the same looking from both the collector and base side. Therefore, the polarity or sign of the current must be the same for both the base and collector otherwise the node rule is violated. There are two other points that I would like to make for the authors: 1.) The gain of a transistor is given by what is gating the transistor. In this case, the base is not gating anything and no current amplification comes from it, the gain that the authors speak of comes from one of the many gates of the transistor (the one that is changing) and they should reserve the terminology for such. Furthermore, the active power dissipation for CMOS comes from the equation $\frac{1}{2} \cdot \text{activity factor} \cdot C \cdot f \cdot V_{dd}^2$. In this case, unless the authors have offered a way reducing the V_{dd} without impacting the other parameters, the parasitic capacitances will dramatically increase the power consumption of the circuit. The main active power consumption is not the joule dissipation of current as the author has stated in his response to the reviewers. I believe that the authors must address the first point made in this review as it violates a key law of physics and the paper should not be accepted until the authors can either explain why it does not apply to them or until they make measurements which are consistent with them.

Responses to the comments of Reviewer #3

Thank you again for your valuable comments. According to your comments, we revised the manuscript. The revised parts are indicated by red characters in the manuscript. The following is the response to your comments.

I thank the authors who have put out the effort to answer the questions of the reviewers. With the additional clarifications, it is clearer now than ever that there is something strange going on. Figure 3d shows that in aspirator mode, there is current flowing from the base which is grounded to the collector which is also grounded. This to me signifies that there is something going on where one or both of the terminals is not truly held at ground.

As we have stated in the original manuscript (in the Methods section), special care has been taken for the offset of the input voltages, or unintentional built-in voltages at the grounded terminals. We have confirmed that no voltage was generated between the two grounded terminals (base and collector). We will explain what we did for this confirmation.

Figure T1(a) shows the voltage setup for the measurement of the current flowing between the base and the collector. The emitter gate was set in OFF state by applying a negative voltage ($V_{EG} = -1.5$ V). Then, in order to avoid any leakage current from the emitter terminal, it was constant-current biased at $I_E = 0$ A. On the other hand, the collector gate was set in ON state by applying a positive voltage ($V_{CG} = 1.5$ V). This voltage is large enough to make the channel beneath the collector gate being in strong inversion. The upper gate voltage was set at $V_{UG} = 3.5$ V, as in the case of the aspirator experiments. The above voltage setup enables us to measure the current flowing between the base and collector without disturbance by the emitter current.

Figure T1(b) shows the base and collector currents, I_B and I_C , measured at 8 K, as a function of time with the base and collector voltages, V_B and V_C , both set at 0 V. One can see that both I_B and I_C have no offset and the fluctuation is about ± 200 pA. This current fluctuation is much smaller than the typical value of the current used for the aspirator experiments, which is on the order of 10 nA.

Fig. T1: Voltage setup (a) and the experimental results (b) for measuring the offset and fluctuation of the base and collector currents, I_B and I_C , when both terminals are grounded. The measurement temperature is 8 K.

Figure T2(a) shows the voltage setup for the second type of measurements. In these measurements, we swept the base voltage V_B keeping the collector grounded ($V_C = 0$ V), and measured the I_B and I_C characteristics. We performed 60 scans in order to check the fluctuation, and the results are shown in Fig. T2(b). One can see that the conduction is ohmic in the measured voltage range, and the resistance was estimated to be about $3 \times 10^4 \Omega$. Figure T2(c) shows the magnified view of Fig. T2(b) around $V_B = 0$ V. One can see in the upper panel that the current fluctuation of ± 250 pA corresponds to the voltage fluctuation of about $\pm 8 \mu\text{V}$. In other words, the voltage difference between the base and collector is zero with an uncertainty only of the order of 10^{-5} V.

From the above results, we conclude that, with both the base and the collector terminals grounded, the voltage difference between these terminals is negligibly small, and no unintentional voltage is generated between the base and the collector. Note that we always confirmed the above conditions being held before taking the data for the manuscript.

For explaining the above experiments, we added some more words in the Methods section in the revised manuscript, and the experimental data (Figs. T1 and T2) were added in the supplemental material.

Fig. T2: Voltage setup (a), the experimental results (b) and the magnified view around $V_B = 0$ V (c) for evaluating the offset and the fluctuation of the base and collector voltages. The measurement temperature is 8 K.

In figure 5, shows that the current to the base and collector are of opposite polarities giving the authors the “amplification” that they speak of in this paper. I would like to point out that if both the base and collector are sitting at ground, then the polarities of these currents must be the same otherwise the node rule of Kirchhoff’s current laws is violated. The common node between the base and collector is the intersection point of the T where the potential is the same looking from both the collector and base side. Therefore, the polarity or sign of the current must be the same for both the base and collector otherwise the node rule is violated.

Considering from the Reviewer’s statement “the polarity or sign of the current must be the same for both the base and collector”, we are afraid that the Reviewer’s concern may not be the violation of the Kirchhoff’s current law, but related to the Kirchhoff’s voltage law, or more specifically, to the positive nature of the resistance. In fact, in the aspirator-mode operation, the symmetry of the potential profile between the base and the collector is broken, and the pumping effect works, which enables electrons to flow against the reverse bias, making the effective resistance negative. In the following, we will first explain that the Kirchhoff’s current law is not violated in the present device. Then, we will explain why the potential symmetry is broken, or why the pumping effect leading to the negative resistance emerges.

The Kirchhoff’s current law (the Kirchhoff’s first law or Kirchhoff’s point rule) states that “the sum of currents in a network of conductors meeting at a point is zero”. This law is the consequence of the charge conservation law.

In order to avoid any confusion and misunderstanding, we first remind the definition of the polarity of the current. In the present manuscript, the polarity of the current is defined by the direction of the electron flow in terms of the terminal, i.e., we define the current to be positive (negative) if electrons flow in (out of) the corresponding terminal. The reason we took this definition is that, with this definition, the above statement for the Kirchhoff’s current law stands.

Fig. T3: Emitter, collector, and base currents, I_E , I_C , and I_B (upper panel) and the sum of the three current (lower panels) for the aspirator-mode (a) and transistor-mode (b) operations. The measurements were performed at 8 K with the substrate and upper-gate voltages, V_{SG} and V_{UG} , of 0 and 3.5 V, respectively.

Then, we claim that, at the intersection of the T branch of the present device, $I_E + I_C + I_B = 0$ holds, where I_E , I_C , and I_B are the emitter, collector, and base currents, respectively. We show all the three currents, I_E , I_C , and

I_B in the aspirator- and transistor-mode operations in the upper panels of Figs. T3(a) and T3(b), which correspond to Fig. 5 in the manuscript. (In Fig. 5, only I_C and I_B are shown.) The lower panels show the sum of the currents, $I_E + I_C + I_B$. One can see that the Kirchhoff's current law is satisfied in both cases.

What makes the present device unusual is not the violation of the Kirchhoff's current law, but the *negative resistance* of the collector channel. We understand the Reviewer's concern in the following way.

Assuming that the base and collector terminals have the same voltages V_0 , and that the voltage at the intersection of the T-branch is V_{TB} , the voltage difference between the terminals and the intersection is $V_0 - V_{TB}$ for both terminals. In such a case, the currents are given by $I_B = R_B(V_0 - V_{TB})$ and $I_C = R_C(V_0 - V_{TB})$, where R_B and R_C are the resistance of the base-intersection and collector-intersection paths, respectively. Then, electrons should move from the T-branch to each terminal if $V_0 - V_{TB} > 0$, or they move from each terminal to the T-branch if $V_0 - V_{TB} < 0$. Figure T4 shows these situations, where the profile of the electrostatic potential ϕ for electrons is drawn. Here, ϕ is defined as $\phi = (-e)V$, where V is the voltage and $e (> 0)$ is the elementary charge. One can see that, in either case, the current polarity is the same, and the potential profile is symmetric with respect to the base and the collector terminals.

Fig. T4: Electron flow for the case where the collector and the base resistances are both positive.

However, this is true only when the polarity of both resistances is positive. In the present device, the resistance of the collector channel R_C becomes negative in the aspirator operation. Figure T5 shows the expected profile of the electrostatic potential ϕ along the path between the base and the collector via the T-branch. (Potential profile in the emitter channel is also shown by the dotted curve.)

Figure 5(a) shows the case where the emitter voltage V_E is small. Nothing special happens and the current flows in a normal way like that shown in Fig. T4(left). That is, electrons from the emitter are drained out either to the base or to the collector. The situation, however, drastically changes when V_E becomes negatively large. Figure 5(b) shows this situation.

Starting from the grounded base terminal, the potential ϕ first gradually decreases (due to a finite value of the base-channel resistance). At the intersection of the T-branch, we have a potential pocket, and then, the potential steeply increases underneath the collector gate, or at the exit of the T-branch to the collector. Finally, the potential again gradually decreases due to a finite value of the resistance of the collector channel, and turns back to zero.

As we have discussed in the manuscript, the potential pocket at the T-branch region is caused by the deficiency of electrons (or accumulation of positive charges) due to the electron-electron ($e-e$) scattering. Electrons coming from the base gain the momentum and energy at the T-branch due to the $e-e$ scattering, which allows these electrons to surmount the potential hump formed beneath the collector gate. (As we have explained in the first Response Letter, the collector gate is not necessary to be biased, or it can be grounded.)

As one can see in the figure, electrons are transferred from the T-branch region to the collector *against the reversed bias* (indicated by the red dashed line in Fig. T5(b)), which in effect is the pumping operation. Therefore, the device can be called the jet pump (just as in the case that the common water aspirator is sometimes called the water jet pump). The energy for the pumping operation is supplied from the high-energy electrons injected from the emitter.

We here emphasize that the pumping effect at the T-branch region is equivalent to putting a battery, or a voltage source there (whose energy is supplied from the emitter electrons), and this is why we can obtain the

electron flow between the two grounded terminals. Therefore, the observed phenomenon does not violate the Kirchhoff's voltage law (or the Kirchhoff's second law) as well.

Precisely speaking, we cannot define the resistance in the T-branch-collector path, because I_C does not change linearly with $V_0 - V_{TB}$ there. However, we can still define the *effective* resistance R_C in the form of $I_C = R_C(V_0 - V_{TB})$, and based on this form, R_C becomes negative.

The origin of the asymmetric potential profile is the *directional* transfer of the emitter-electron momentum towards the collector. This is why the potential along the collector channel is always higher than that along the base channel.

Fig. T5: Potential profile and electron flow in the device when the collector voltage V_E is small (a) and when V_{EG} is negatively large (b).

We should mention that this is not a new phenomenon. Similar current behavior has already been reported in GaAs/AlGaAs (micrometer scaled) devices listed below, and similar explanation has been given there. (Note that the significance of the present work is not the finding of a new phenomenon, but the demonstration that the phenomenon can be observed even in nanometer scale, and can be controlled in Si MOS devices.)

[1] Kaya, I. I. & Eberl, K.

Absolute negative resistance induced by directional electron-electron scattering in a two-dimensional electron gas. Phys. Rev. Lett. 98, 186801_1-4 (2007).

[2] Taubert, D. et al.

An electron jet pump: The Venturi effect of a Fermi liquid. J. Appl. Phys. 109, 102412_1-5 (2011).

In summary, in the present device, the Kirchhoff's current law is not violated. The asymmetric potential profile is due to the directional momentum transfer towards the collector, and the current flow between the two grounded terminals is due to the pumping effect, which makes the device an aspirator, or jet pump.

In order to show that the Kirchhoff's current law is satisfied in the present measurements, we added Fig. T3 in the supplemental material as Fig. S2. We also added the explanation about the pumping operation in page 6 in the revised manuscript. Note that we have already mentioned the negative resistance in the manuscript (underlined part in page 6).

There are two other points that I would like to make for the authors:

We appreciate such fundamental comments.

1.) The gain of a transistor is given by what is gating the transistor. In this case, the base is not gating anything and no current amplification comes from it, the gain that the authors speak of comes from one of the many gates of the transistor (the one that is changing) and they should reserve the terminology for such.

We are not sure if we understand the Reviewer's requests correctly, but the Reviewer might ask us to answer which gate is responsible for the collector-current enhancement and, according to this assignment, to use the terminology (such as gain, gating, and amplification) in an appropriate and correct way.

It is the collector gate that controls the enhancement factor $R_1 = |I_C/I_E|$. However, we did not use the terms "gain" in the manuscript for describing the role of the collector gate. This is because the way of the enhancement of the collector current is different from the "current gain" (in bipolar transistors), considering its common definition. The collector current increases not because the emitter current itself is amplified, but merely because the base current is added due to the $e-e$ scattering effect, as explained in Fig. T5. However, we used the term "amplification". We are afraid that this term suggests the "current gain". Therefore, we decline to use the term "amplification" in order to avoid confusion. In the following, we will explain the device operation of the present device in order to make our claim clear.

We first show, in Fig. T6, the $R_1 (= |I_C/I_E|)$ as a function of the collector-gate voltage, V_{CG} , with the substrate terminal grounded, or $V_{SG} = 0$ V. (This figure corresponds to Fig. 1(e) in the manuscript, where we set the substrate voltage $V_{SG} = -15$ V.) As one can see, the R_1 is a function of V_{CG} , or the collector gate controls the R_1 . This is because the collector gate controls the effect of the $e-e$ scattering (by which the base current is generated) by changing the height of the potential hump shown in Fig. T5. We mention that the enhancement of the collector current I_C is due to the electron flow from the grounded base terminal, not due to the amplification of the emitter current I_E (I_E is kept at -10 nA).

Fig. T6: $R_1 (= |I_C/I_E|)$ as a function of the collector-gate voltage V_{CG} with the substrate terminal grounded. The upper-gate voltage V_{UG} is fixed at 3.5 V.

We next explain the device operation. We mention that the present device is basically a MOSFET. In MOSFETs, what is gating the device is the gate terminal, which controls the trans-conductance, $G_m = I_{SD}/V_G$, where I_{SD} and V_G are the current flowing between the source and drain, and the gate voltage, respectively.

As shown in Fig. 5 in the manuscript (or in Fig. T3 in this Response Letter), the device is operated as a function of emitter-gate voltage, V_{EG} . Therefore, we can say that the emitter gate is gating the device (as a MOSFET), or the emitter gate controls the G_m of the present device. (More specifically, the emitter gate controls the emitter current I_E and the G_m of the emitter channel.)

In this MOSFET operation, the role of the collector gate is to control the effect of the $e-e$ scattering. By setting the collector-gate voltage V_{CG} to an appropriate value (which is simply to make the collector gate grounded), we can maximize the effect and can induce the electron flow from the grounded base terminal. This

base current is added to the emitter current, resulting in the enhancement of the collector current. This can be regarded as the increase in the effective trans-conductance of the device.

Because the present device has three current terminals and the base is grounded, the above device operation can be compared to the base-common operation of the bipolar transistor, which is characterized by $\alpha = I_C/I_E$. The significance of the present device is that, contrary to the ordinary bipolar transistor, we can make α (or R_I in the present definition) larger than unity. This unusual behavior is because, by enjoying the $e-e$ scattering, we can efficiently use the energy of the emitter (or source) electrons, which otherwise would be dissipated to the phonon bath, resulting in the pumping operation (Fig. T5(b)).

As we have explained above, in the present device, the collector (or drain) current is increased not because of the amplification of the emitter (source) current itself, but merely because of the addition of the base current owing to the $e-e$ scattering.

In summary, we have explained the origin of the collector-current enhancement, and mentioned that we did not use the term “gain” for describing the role of the collector gate. However, we used the term “amplification”. We are afraid that this term suggests the “current gain” in the bipolar transistor. Therefore, we decline to use the term “amplification” in order to avoid confusion. Instead, we use the terms “enhancement” or “increase in the collector current”. We hope we adequately address the Reviewer’s comments and requests.

Furthermore, the active power dissipation for CMOS comes from the equation $1/2 \cdot \text{activity factor} \cdot C \cdot f \cdot V_{dd}^2$. In this case, unless the authors have offered a way reducing the V_{dd} without impacting the other parameters, the parasitic capacitances will dramatically increase the power consumption of the circuit. The main active power consumption is not the joule dissipation of current as the author has stated in his response to the reviewers.

The feature of the present device is that it can enjoy the $e-e$ scattering, and enables us to use the energy otherwise to be dissipated to the phonon bath, which results in the increase of the output (collector) current, making the device faster than the conventional MOSFETs. This feature is important for the energy-consumption issue of the electronic circuit. We will explain this point in the following.

As the Reviewer points out, the dynamic power dissipation of the CMOS circuits is uniquely determined by the operation frequency f , the power supply voltage V_{dd} , and the capacitance C (including the parasitic component) of the output node. Therefore, even if we operate the device in the aspirator mode, the energy consumption is unchanged.

What the present device reduces (or improves) is not the power consumption itself, but the energy-delay product $E\tau$, where $E = (1/2)CV_{dd}^2$ is the energy consumption in one cycle of the charging/discharging process, and τ is the time required for charging the capacitance C . This is owing to the fact that we can generate larger current (collector current) for charging the output node. In other words, the drivability can be effectively increased by the aspirator operation, which enables us to make a faster charging of the output node.

Upon the enhancement of the collector current by R_I , the delay time is reduced by $1/R_I$. This means that we can increase the operation frequency f by R_I . Alternatively, if we design the circuit so that the operation frequency f is unchanged (or the drive current unchanged), then, we can reduce the power supply voltage V_{dd} by $1/R_I$ (under the most ideal conditions), which leads to the reduction of the energy consumption by $(1/R_I)^2$. This is why we explained, in the first Response Letter, that “the device is energy efficient”.

We mention that the origin of the active power consumption in CMOS circuits is the Joule dissipation of the current, where the current means the electron flow in the channel and in the wire for charging the output node. $E = (1/2)CV_{dd}^2$ is the electrostatic charging energy that the capacitor (accumulating the charge $Q (= CV_{dd})$) holds. During one cycle of the charging and discharging of the capacitor, the same amount of the energy is dissipated by the Joule heating due to a finite value of the resistance. In other words, in CMOS circuits, the charging energy $E = (1/2)CV_{dd}^2$ is unavoidably wasted due to the Joule dissipation during the operation.

What we have demonstrated here is that, by enjoying the $e-e$ scattering, we can efficiently use the energy otherwise to be dissipated by the Joule heating, which leads to the increase in the drivability, or to making the device faster, by which we expect either a faster or lower-energy circuit to be realized.

We are afraid, however, that the term “energy efficient” is misleading because it suggests the reduction of the energy consumption itself. Therefore, we deleted this term from the manuscript. Instead, we used “the increase in the drivability” for characterizing the merit of the device (in page 3 and 11 in the revised manuscript). Accordingly, we revised the abstract and the summary as well.

I believe that the authors must address the first point made in this review as it violates a key law of physics and the paper should not be accepted until the authors can either explain why it does not apply to them or until they make measurements which are consistent with them.

As we have explained, the Kirchhoff’s current law is not violated in the present device.
It would be appreciated if you could accept our response and consider the acceptance to the publication.
Thank you for your kind consideration.

REVIEWERS' COMMENTS:

Reviewer #1 (Remarks to the Author):

The authors, in my opinion, carefully addressed the issues that were raised by Reviewer 3, especially what seemed to be a violation of a fundamental physical rule. Nevertheless, I would strongly encourage them to include Figures T4 and T5 as well as the accompanying explanations into the supplemental material of the paper. They represent a very nice explanation of what happens inside their device and can therefore be useful to many readers.

Reviewer #2 (Remarks to the Author):

I found that the reply from the authors is convincing and the revision is adequate. I judge that the manuscript is publishable.

Reviewer #3 (Remarks to the Author):

Upon looking at the citations, it does seem that the physics does match with the previous report, but instead in silicon. I will recommend the work for publication, but would like to point out that I continue to believe that the authors are overstating the importance of their work. If one disregards the fact that this effect is only currently being demonstrated at low temperatures, there are still multiple things which do not make this a usable device for switching. First, the device is operating in the subthreshold regime with respect to the collector gate meaning that the drive current to the collector will be particularly low and the delay times will be high and therefore it will not switch quickly. Also, as mentioned previously the device requires high voltage (on the emitter//collector terminal) which means the device will have high values of $\frac{1}{2} CV^2$. To put it simply, one can also look at the power dissipation as $Q \cdot V$ and if the authors' beliefs are true on using scattering as a means of increasing the number of carriers, then the amount of charge necessary will be high and the voltage requirements will also be high. I am also unsure how well the device will behave as a switch in the regime that the authors are operating the device. Finally, it will be extremely difficult to make the footprint of this device to be small (think additional vias etc) and therefore it will unlikely be used for any logic application as the author states.

Responses to the Reviewers' comments

Reply to Reviewer #1:

The authors, in my opinion, carefully addressed the issues that were raised by Reviewer 3, especially what seemed to be a violation of a fundamental physical rule. Nevertheless, I would strongly encourage them to include Figures T4 and T5 as well as the accompanying explanations into the supplemental material of the paper. They represent a very nice explanation of what happens inside their device and can therefore be useful to many readers.

According to your suggestion, we included Figs. T4 and T5 in the second response letter as Supplementary Figures 2 and 3 in Supplementary Note 2, together with the accompanying explanation.

We appreciate your reviewing and many valuable comments.

Reply to Reviewer #2:

I found that the reply from the authors is convincing and the revision is adequate. I judge that the manuscript is publishable.

We appreciate your reviewing and many valuable comments.

Reply to Reviewer #3:

Thank you again for your valuable comments. The following is the response to your comments. We will first answer your comments one by one, and then mention the key points to be studied in the future.

Upon looking at the citations, it does seem that the physics does match with the previous report, but instead in silicon. I will recommend the work for publication, but would like to point out that I continue to believe that the authors are overstating the importance of their work. If one disregards the fact that this effect is only currently being demonstrated at low temperatures, there are still multiple things which do not make this a usable device for switching. First, the device is operating in the subthreshold regime with respect to the collector gate meaning that the drive current to the collector will be particularly low and the delay times will be high and therefore it will not switch quickly.

We first remind that, although most of the data (Figs. 1 – 4 in the manuscript) were taken as a function of the collector gate V_{CG} , this is for clarifying the physics behind the device operation. In the actual device operation, however, we use the emitter gate as a principal gate, while the collector-gate voltage V_{CG} is fixed at a constant value, as shown in Fig. 5 in the manuscript.

Then, what determines the level of the injection current is not the collector gate but the emitter gate, and the device operates in the inversion regime of the emitter gate. In other words, the level of the current before the enhancement by the $e-e$ scattering is dominated by the voltage to the emitter gate V_{EG} , and the V_{EG} can be larger than its threshold voltage. In the aspirator mode, the current level is further increased by the addition of the base current due to the $e-e$ scattering. Therefore, the device can operate faster than that in the inversion regime of conventional MOSFETs.

In the present device, the collector gate was set at around the boundary between the subthreshold and weak inversion regimes to maximize the current enhancement. We emphasize that this setting does not result in the current reduction because, due to the $e-e$ scattering, electrons in the T-branch region have gained energy large enough to surmount the potential hump created by the collector gate. In other words, at the T-branch region, energy distribution of electrons is far from the equilibrium Fermi-Dirac distribution.

Also, as mentioned previously the device requires high voltage (on the emitter//collector terminal) which means the device will have high values of $\frac{1}{2} CV^2$.

The set value of the voltage between the emitter and collector terminals, or $|V_E|$ in our definition, is determined by the requirement of how fast the device should operate. Figure F1, which is the linear-scale plot of Fig. 3b in the manuscript, shows the $R_I (= |I_C/I_E|$ with I_C and I_E respectively the collector and emitter currents) as a function of $|V_E|$. As one can see, higher R_I requires higher $|V_E|$, but it is still larger than unity, e.g., $R_I \approx 2$ at $|V_E| \approx 0.6$ V for $V_{CG} = 0.36$ V. (See the red curve in the figure. Note that the collector gate was not grounded but set to $V_{CG} = 0.36$ V because of the biasing to the substrate, $V_{SG} = -15$ V.) The value $|V_E| \approx 0.6$ V is moderately low comparing to the power supply voltage of current CMOS circuits.

From the above discussion, one can understand that the lowest level at which $|V_E|$ can be set is dependent on how high the value of R_I (with a fixed $|V_E|$) can be obtained. Maximum achievable value for R_I (with a fixed $|V_E|$) will be dependent on the size and shape of the device and also on the strength of the momentum-nonconserving scatterings, such as interface-roughness and phonon scatterings, which are competing scattering processes against the $e-e$ scattering. At present, maximum achievable value of R_I is unclear, and clarifying it remains as a future work. However, as we will show at the end of this Response Letter, we expect that R_I can exceed 3 even for $|V_E| = 0.3$ V at room temperature, for the optimum case.

Note that, as we have explained to Reviewer #2 in the first Response Letter, the increase in the enhancement factor, $R_I - R_I(V_E=0)$ is expected to be proportional to the forward momentum p of an injected electron, which is given by $p = (2m(|eV_E| + E_F))^{1/2} \approx (2m|eV_E|)^{1/2}$ (for $|eV_E| \gg E_F$). That is, $R_I - R_I(V_E=0) \propto |V_E|^{1/2}$. The observed dependence of R_I on $|V_E|$ can be explained by this simple formula.

Fig. F1. Dependence of R_I on $|V_E|$. The arrow shows the accidental drop of R_I .

To put it simply, one can also look at the power dissipation as $Q \cdot V$ and if the authors' beliefs are true on using scattering as a means of increasing the number of carriers, then the amount of charge necessary will be high and the voltage requirements will also be high.

We understand the Reviewer's question in the following way. Addition of the base terminal results in the increase in the capacitance to be charged, which increases the energy consumption, and destroys or compensates the merit of the high drivability of the aspirator. As you point out, this effect will need to be carefully considered for circuit design in the future. In the following, we will briefly explain our present understanding on this point.

As we have discussed in the second Response Letter, what the present device reduces (or improves) is not the power consumption itself, but the energy-delay product $E\tau$, where $E = (1/2)CV_{dd}^2$ is the energy consumption in one cycle of the charging/discharging process, and τ is the time required for charging the capacitance C . This is owing to the fact that we can generate larger current (collector current) for charging the output node. Upon the enhancement of the collector current by R_I , the delay time is reduced by $1/R_I$. This means that we can increase the operation frequency f by R_I . Alternatively, if we design the circuit so that the operation frequency f is unchanged (or the drive current is unchanged), then we can reduce the power supply voltage V_{dd} by $1/R_I$ (under the most ideal conditions), which leads to the reduction of the energy consumption by $(1/R_I)^2$.

From the above argument, one can see that, considering the circuit operation with a fixed f , the energy consumption can be reduced as $(1/R_I)^2$, while it increases only linearly with C . Therefore, as long as $R_I \gg 1$ (e.g., $R_I \gtrsim 3$ as in the present device) and the base-node capacitance C_B is not too large ($C_B \lesssim C_{OUT}$, where C_{OUT} is the output-node capacitance), we can reduce the energy consumption of the circuit.

Note that, as shown in Fig. F1, the R_I is decreased with decreasing $|V_E|$. However, its dependence is weak, scaled only as $|V_E|^{1/2}$.

I am also unsure how well the device will behave as a switch in the regime that the authors are operating the device.

We understand that Reviewer #3 asks us the subthreshold slope (allowable minimum value of the voltage swing) and the delay time (the time required for changing the states from 0 to 1 and vice versa).

The subthreshold slope in the aspirator mode is the same as that in the transistor mode, that is, the same as that of conventional MOSFETs. This is because the R_I is, in principle, independent of the level of the injection (source) current.

The delay time is determined by the RC time constant of the base node. As explained in the second Response Letter and in the revised supplemental material (Supplementary Figure 9), the base-collector resistance R is on the order of $10^4 \Omega$ in the present device. Assuming a base-node capacitance C_B of 1 fF, which is a typical value of the load capacitance for one logic gate, the RC time constant comes to 10 ps, corresponding to a frequency of 100 GHz. These values indicate that the device works sufficiently fast for circuit applications. By making the base node smaller, the RC time becomes shorter, and the device responds faster.

Finally, it will be extremely difficult to make the footprint of this device to be small (think additional vias etc) and therefore it will unlikely be used for any logic application as the author states.

We understand that Reviewer #3 asks us about the lithographic issue. As we have mentioned in the manuscript and explained in detail in the supplemental material, the device size should be less than 20 – 30 nm for room-temperature operation. This means that the Si-wire width, the gate length, and the spacing between the gates should be designed with the same order dimension.

As you point out, the fabrication of circuits composed of the present device will be more difficult than those of the conventional MOS transistors. Therefore, efforts for simplifying the device structure, e.g., exploring the possibility of eliminating the collector gate, have to be pursued in the future. Nevertheless, we anticipate that the fabrication of the present device is possible to achieve.

As you may know, according to the 2017 edition of the IRDS (International Roadmap for Devices and Systems: <https://irds.ieee.org/roadmap-2017>, See “More Moore”), MPU/SoC Metal 1/2 Pitch has already reached 18 nm in 2017, and is expected to reach 12 and 7 nm in 2021 and 2027, respectively. Physical Gate Length for HP Logic has already reached 20 nm in 2017, and is expected to reach 16 and 12 nm in 2021 and 2027, respectively. In addition, EUV/DSA (Extreme Ultraviolet/Directed Self-Assembly) lithography and 3D stacked structure will be introduced in 2024 and 2030, respectively, which are promising for increasing the flexibility of the device structure. All these prospects are promising for the miniaturization of the present device.

In summary, the comments raised by Reviewer #3 reveal that there are several points to be studied in the future. Among them, how high the R_I can be achieved is the most fundamental for the present device, and needs to be clarified with the first priority. Hereafter, we will explain briefly our estimation of the maximum achievable value of R_I .

In the saturation regime of a scaled MOSFET, the level of the current is determined by the source injection current (Refs. 4 and 5), or by the forward momentum of an incident electron, p_{IN} . It can be expressed in a simple form as $p_{IN} = (2mkT)^{1/2}$ at room temperature, where m , k , and T are the effective mass, the Boltzmann constant, and the absolute temperature, respectively. (At low temperature, it is expressed as $p_{IN} = (2mE_F)^{1/2}$, where E_F is the Fermi energy.) The incident electron is accelerated in the channel and the momentum reaches $p = (2m(kT + eV))^{1/2}$. However, the additional momentum gained by the electric field $p_{EL} \approx (2meV)^{1/2}$ (for $p_{EL} \gg p_{IN}$) is unavoidably wasted to the phonon bath in the MOSFET. The aspirator, on the other hand, uses it for generating the base current. If we could take a full use of it, the R_I will reach $R_I \sim p / p_{IN} \approx (eV/kT)^{1/2}$, or $R_I \approx 3.5$ for $V = 0.3$ V at room temperature. In addition, if the injected electrons could scatter selectively with cold electrons with a forward momentum, as Kaya and Eberl claimed (ref. 11), further increase in R_I may be possible.

The above estimation is encouraging, but more detailed analysis about the dependence of R_I on the device structure and the influence of the momentum-nonconserving scattering is called for to make any decisive conclusions. Exploring the circuit architectures suitable for the device will also be needed.

We added these points from line 10 to line 23 of page 14 in the revised manuscript (converted PDF).

It is appreciated if you could accept our response. Also, we thank you for your reviewing and many valuable comments.